# Exaptation of ancestral cell-identity networks enables C$_4$ photosynthesis

Joseph Swift[1,6], Leonie H. Luginbuehl[2,6 ✉], Lei Hua[2], Tina B. Schreier[2,5], Ruth M. Donald[2], Susan Stanley[2], Na Wang[2], Travis A. Lee[1], Joseph R. Nery[3], Joseph R. Ecker[1,3,4 ✉] & Julian M. Hibberd[2 ✉]

C$_4$ photosynthesis is used by the most productive plants on the planet, and compared with the ancestral C$_3$ pathway, it confers a 50% increase in efficiency[1]. In more than 60 C$_4$ lineages, CO$_2$ fixation is compartmentalized between tissues, and bundle-sheath cells become photosynthetically activated[2]. How the bundle sheath acquires this alternate identity that allows efficient photosynthesis is unclear. Here we show that changes to bundle-sheath gene expression in C$_4$ leaves are associated with the gain of a pre-existing *cis*-code found in the C$_3$ leaf. From single-nucleus gene-expression and chromatin-accessibility atlases, we uncover DNA binding with one finger (DOF) motifs that define bundle-sheath identity in the major crops C$_3$ rice and C$_4$ sorghum. Photosynthesis genes that are rewired to be strongly expressed in the bundle-sheath cells of C$_4$ sorghum acquire *cis*-elements that are recognized by DOFs. Our findings are consistent with a simple model in which C$_4$ photosynthesis is based on the recruitment of an ancestral *cis*-code associated with bundle-sheath identity. Gain of such elements harnessed a stable patterning of transcription factors between cell types that are found in both C$_3$ and C$_4$ leaves to activate photosynthesis in the bundle sheath. Our findings provide molecular insights into the evolution of the complex C$_4$ pathway, and might also guide the rational engineering of C$_4$ photosynthesis in C$_3$ crops to improve crop productivity and resilience[3,4].

In multicellular systems, changes to the patterning of gene expression drive modifications in cell function and trait evolution. One notable example is found in more than 60 plant lineages, in which compartmentalization of photosynthesis between cell types allowed the evolution of the efficient C$_4$ pathway from the ancestral C$_3$ state[5,6]. In most land plants, CO$_2$ fixation occurs in mesophyll cells and is dependent on ribulose-1,5-bisphosphate carboxylase/oxygenase (RuBisCO). Because the first fixation product of RuBisCO is a three-carbon metabolite, this pathway has been termed C$_3$ photosynthesis[7]. Although most land plants use the C$_3$ pathway, RuBisCO is not able to completely discriminate between CO$_2$ and O$_2$. In addition to the loss of carbon fixation, when RuBisCO carries out oxygenation reactions it generates a toxic intermediate, phosphoglycolate, which must rapidly be metabolized through the energy-intensive photorespiratory cycle[8]. In multiple plant lineages, including staple crops such as maize and sorghum, evolution has reconfigured the functions of mesophyll and bundle-sheath cells such that CO$_2$ fixation by RuBisCO is repressed in the mesophyll and activated in the bundle sheath (Fig. 1a). These species are known as C$_4$ plants because the pathway's first step produces C$_4$ acids in mesophyll cells, which then diffuse into bundle-sheath cells before decarboxylation in close proximity to RuBisCO[9]. This process leads to a tenfold increase in CO$_2$ concentration in bundle-sheath chloroplasts[10], thereby reducing oxygenation reactions so that

photosynthetic as well as water and nitrogen use efficiencies are markedly increased[1]. As a result, C$_4$ plants grow particularly well in hot and dry climates and constitute some of the most productive crop species in the world[11,12].

In both C$_3$ and C$_4$ plants, photosynthetic efficiency is dependent on mechanisms that pattern differential gene expression between each cell type of the leaf. However, so far, only a small number of *cis*-elements that control the cell-specific expression of C$_4$ genes have been identified[13–18], and it is not clear how strict partitioning of photosynthesis between cell types is established and maintained, or how these patterns change to allow the evolution of C$_4$ photosynthesis from the ancestral C$_3$ pathway.

To define the transcriptional identity of each cell type and uncover how the patterning of photosynthesis gene expression is established, we generated gene-expression and chromatin-accessibility atlases in rice and sorghum, which use the C$_3$ and the C$_4$ pathways, respectively. We sampled nuclei after the transfer of seedlings from dark to light—a stimulus that induces photomorphogenesis and thus activates the expression of photosynthesis genes. Both rice and sorghum are models and diploid crops of global importance, representing distinct clades in the monocotyledons that diverged approximately 81 million years ago[19]. Karyotype reconstruction indicates a shared genome structure in both sorghum and rice, as well as their grass ancestor[20]. Thus, molecular signatures of each cell type shared by rice and sorghum may also

[1]Plant Biology Laboratory, Salk Institute for Biological Studies, La Jolla, CA, USA. [2]Department of Plant Sciences, University of Cambridge, Cambridge, UK. [3]Genomic Analysis Laboratory, Salk Institute for Biological Studies, La Jolla, CA, USA. [4]Howard Hughes Medical Institute, Salk Institute for Biological Studies, La Jolla, CA, USA. [5]Present address: Department of Biology, University of Oxford, Oxford, UK. [6]These authors contributed equally: Joseph Swift, Leonie H. Luginbuehl. ✉e-mail: lhl28@cam.ac.uk; ecker@salk.edu; jmh65@cam.ac.uk

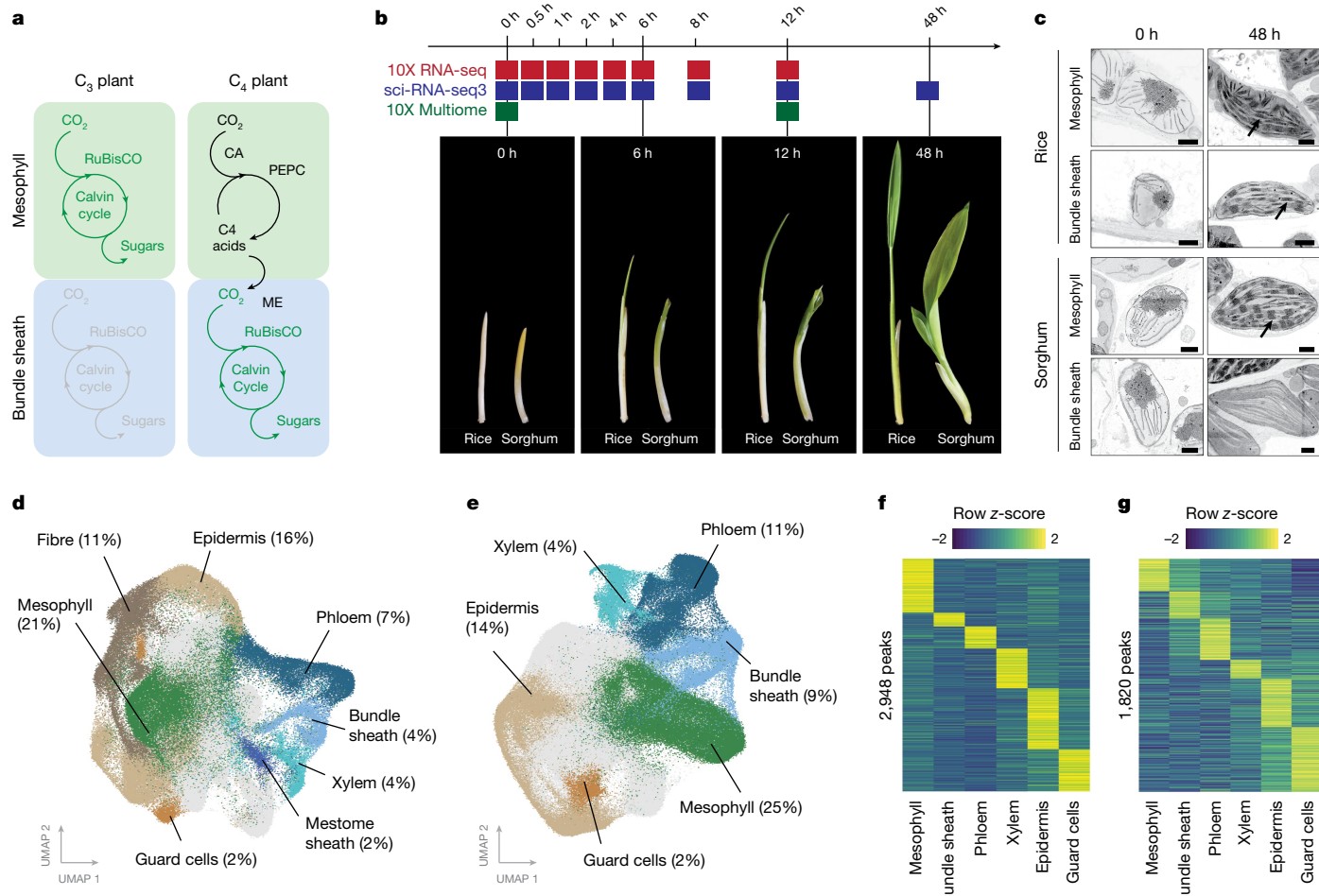

**Fig. 1 | Gene expression and chromatin accessibility of single nuclei from rice and sorghum undergoing de-etiolation. a**, Photosynthesis in mesophyll and bundle-sheath cells of $C_3$ and $C_4$ plants. CA, carbonic anhydrase; ME, malic enzyme; PEPC, phospho*enol*pyruvate carboxylase. **b**, Schematic of de-etiolation time course. Plants were grown in the dark for 5 days (0-h time point) before exposure to light. **c**, Scanning electron micrographs (SEMs) of etioplasts and chloroplasts in the mesophyll and bundle-sheath cells of rice and sorghum. Thylakoid stacks (black arrows) were present in rice mesophyll and bundle-sheath chloroplasts as well as sorghum mesophyll chloroplasts (SEMs consistent across three biological replicates). Scale bars, 1 μm. **d,e**, UMAP of transcript profiles of rice (**d**) and sorghum (**e**) nuclei from all time points assayed. Distinct colours indicate different cell types. **f,g**, In total, 2,948 accessible chromatin peaks were cell-type specific in rice promoters (**f**), and 1,820 peaks were cell-type specific in sorghum promoters (**g**).

be found in the approximately 11,000 species derived from their last common ancestor.

In both species, we found that gene expression was rapidly induced by light in all cell types. However, before the perception of light, the expression and chromatin accessibility of many photosynthesis genes was conditioned by cell identity. Although transcriptional cell identities changed across species, we found that *cis*-elements defining cell identity were conserved. Genes that rewire their expression to become bundle sheath specific in $C_4$ sorghum acquired ancestral *cis*-elements that direct expression in the $C_3$ bundle sheath. The simplest explanation from these findings is that the evolution of photosynthesis involves $C_4$ genes acquiring *cis*-elements associated with bundle-sheath identity, which then harness a stable patterning of transcription factors between cell types of $C_3$ and $C_4$ leaves.

## Rice and sorghum single-nucleus atlases

To understand how different cell types in rice and sorghum shoots respond to light, we grew seedlings of each species in the dark for five days and then exposed them to a light–dark photoperiod for 48 h (Fig. 1b). As expected, shoot tissue underwent photomorphogenesis during this time. Leaves emerged and chlorophyll accumulated within

the first 12 h of de-etiolation (Fig. 1b and Extended Data Fig. 1). Scanning electron microscopy (SEM) showed that etioplasts in both mesophyll and bundle-sheath cells contained prolamellar bodies before light exposure (Fig. 1c and Extended Data Fig. 1). Within 12 h of light, etioplasts had converted into mature chloroplasts with assembled thylakoid membranes. Compared with rice, chloroplast development was more pronounced in the bundle sheath of sorghum, and clear differences in thylakoid stacking in chloroplasts from sorghum mesophyll and bundle-sheath cells were evident (Fig. 1c and Extended Data Fig. 1).

Underlying this cellular remodelling and activation of the photosynthetic apparatus are changes in gene regulation. However, so far, these have been described in bulk tissue samples, so how each cell type responds is not known[21–25]. We generated single-nucleus atlases of transcript abundance for both rice and sorghum shoots as they undergo photomorphogenesis. Shoot tissue at nine time points during de-etiolation was collected and the nuclei were sequenced[26] (Fig. 1b and Extended Data Fig. 1), resulting in gene-expression atlases derived from 190,569 and 265,701 nuclei from rice and sorghum, respectively. We also assayed cell-specific changes in chromatin accessibility at 0 h and 12 h after light exposure by sequencing 22,154 and 20,169 nuclei from rice and sorghum, respectively.

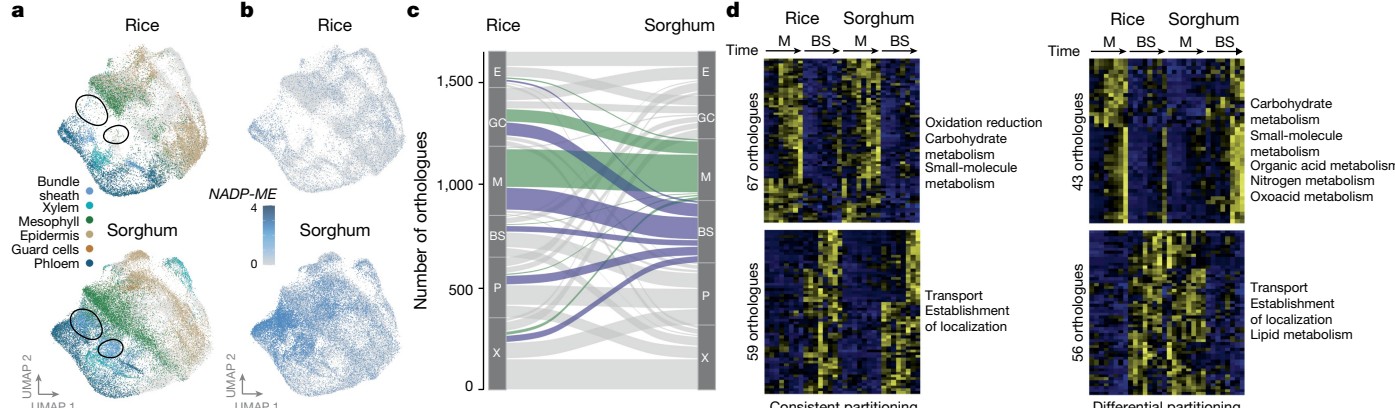

**Fig. 2 | Rice and sorghum bundle-sheath cells show low conservation in transcript partitioning. a**, Pan-transcriptome of rice and sorghum nuclei based on orthologues. UMAPs indicate rice (top) and sorghum (bottom) nuclei. Areas indicated with black circles indicate nuclei from the sorghum bundle sheath that do not co-cluster with rice nuclei. **b**, Transcript abundance for sorghum *NADP-ME* and its rice orthologue displayed in UMAP format from **a**. **c**, Sankey plot of changes in the cell-type partitioning of marker genes. Markers for sorghum mesophyll and bundle-sheath cell types are highlighted in green

and blue, respectively. BS, bundle sheath; E, epidermis; GC, guard cell; M, mesophyll; P, phloem; X, xylem. **d**, Differentially expressed orthologue pairs in rice and sorghum mesophyll and bundle-sheath cells. Genes fall into two categories: those consistently partitioned (more highly expressed in the same cell type in both rice and sorghum); and those differentially partitioned (swapping expression from one cell type to the other). GO terms associated with each category are shown on the right.

These outputs were visualized using uniform manifold approximation projection (UMAP), and 19 distinct clusters were identified for each species (Fig. 1d,e, Extended Data Fig. 2 and Supplementary Table 1). Using the expression of previously described marker genes and their orthologues, cell types were assigned to each main cluster (Extended Data Fig. 2). This included mesophyll, guard, epidermal, xylem parenchyma and phloem cells (Fig. 1d,e). Gene Ontology (GO) terms derived from cluster-specific genes reflected previously documented functions for each cell type (Supplementary Table 1). For example, mesophyll nuclei showed high expression of genes involved in photosynthesis, and clusters containing nuclei from epidermis cells were enriched in genes involved in lipid biosynthesis and export, consistent with the role of this tissue in cutin production[27].

We identified nuclei of the sorghum bundle sheath through the expression of $C_4$ genes such as NADP-malic enzyme (*NADP-ME*) and glycine decarboxylase (*GDC*) (Extended Data Fig. 2). However, to our knowledge there are no such markers for the bundle sheath in rice undergoing photomorphogenesis. We therefore generated a stable reporter line in which bundle-sheath nuclei were labelled with an mTurquoise2 reporter under the control of the *PHOSPHOENOLCARBOXYKINASE* promoter from the $C_4$ plant *Zoysia japonica* (Extended Data Fig. 3). This promoter drives bundle-sheath expression in rice[27]. Transcriptome sequencing of a rice leaf nuclei population enriched with bundle-sheath nuclei identified 14 clusters, with the expression of mTurquoise2 identifying the rice bundle sheath (Extended Data Fig. 3). In the same cluster, we also detected the expression of genes such as *PLASMA MEMBRANE INTRINSIC PROTEIN* (*PIP1.1*) and *SULFITE REDUCTASE* (*SIR*), which have previously been shown to be expressed in the bundle sheath of mature rice leaves[28] (Supplementary Table 2). Using marker genes from this cluster, it was then possible to annotate nuclei with bundle-sheath identity in our de-etiolation dataset (Extended Data Fig. 3).

Complementing this atlas describing cell-type gene expression, the multiome assay (RNA sequencing (RNA-seq) and assay for transposase-accessible chromatin with sequencing (ATAC-seq)) allowed changes in chromatin accessibility during photomorphogenesis to be detected. After cross-validation with single-nucleus transcriptional atlases, multiome atlases identified six cell types from each species (Extended Data Fig. 4) and revealed accessible peaks in promoter regions of genes specific to each cell type (Fig. 1f,g). As expected, genes proximal to these peaks held enriched GO terms associated with

known cell functions (Supplementary Table 3). Peaks were detected upstream of canonical marker genes for each cell type. For example, the promoters of the RuBisCO small subunit (*OsRBCS4*) from rice and *NADP-ME* from sorghum were most accessible in mesophyll and in bundle-sheath cells, respectively (Extended Data Fig. 4). In summary, these data provide cell-level insights into changes in gene expression and chromatin accessibility associated with the induction of photosynthesis in cereal crops.

## Repurposing the bundle sheath

$C_4$ evolution has repeatedly repurposed the bundle sheath to perform photosynthesis[29,30]. However, the extent to which this cell type has been altered transcriptionally is not known. To understand how the transcriptional identities of each cell type from rice and sorghum differ, we generated a pan-transcriptome atlas of photosynthetic tissue sampled at 48 h after light exposure. This was achieved by identifying sorghum and rice orthologues and clustering nuclei from both species together. Despite the evolutionary distance between rice and sorghum, most cell types from these species co-clustered (Fig. 2a). The most notable exception were nuclei from bundle-sheath cells, with those from the sorghum bundle sheath clustering distinctly from those from the rice bundle sheath (Fig. 2a and Extended Data Fig. 5). Supporting this observation, GO enrichment analysis indicated that $C_3$ and $C_4$ bundle-sheath cells have distinct functions—whereas genes expressed in the bundle sheath of rice were predominantly associated with transport and localization, those of sorghum were associated with organic acid metabolism and the generation of precursor metabolites and energy (Supplementary Table 1).

More than 180 genes specific to the bundle-sheath cells of sorghum had orthologues that were either poorly expressed or not specific to any cell type in rice (Supplementary Table 4). For example, the canonical $C_4$ gene *NADP-ME* was strongly and specifically expressed in the sorghum bundle sheath but was poorly expressed and not cell specific in rice (Fig. 2b). Similar patterns of high and localized expression were evident in the sorghum bundle sheath for other genes involved in photosynthesis, photorespiration and chloroplast functions (Extended Data Fig. 5 and Supplementary Table 4). The bundle sheath of $C_4$ sorghum also lost the expression of genes associated with this cell type in rice (Extended Data Fig. 5 and Supplementary Table 4). Notably,

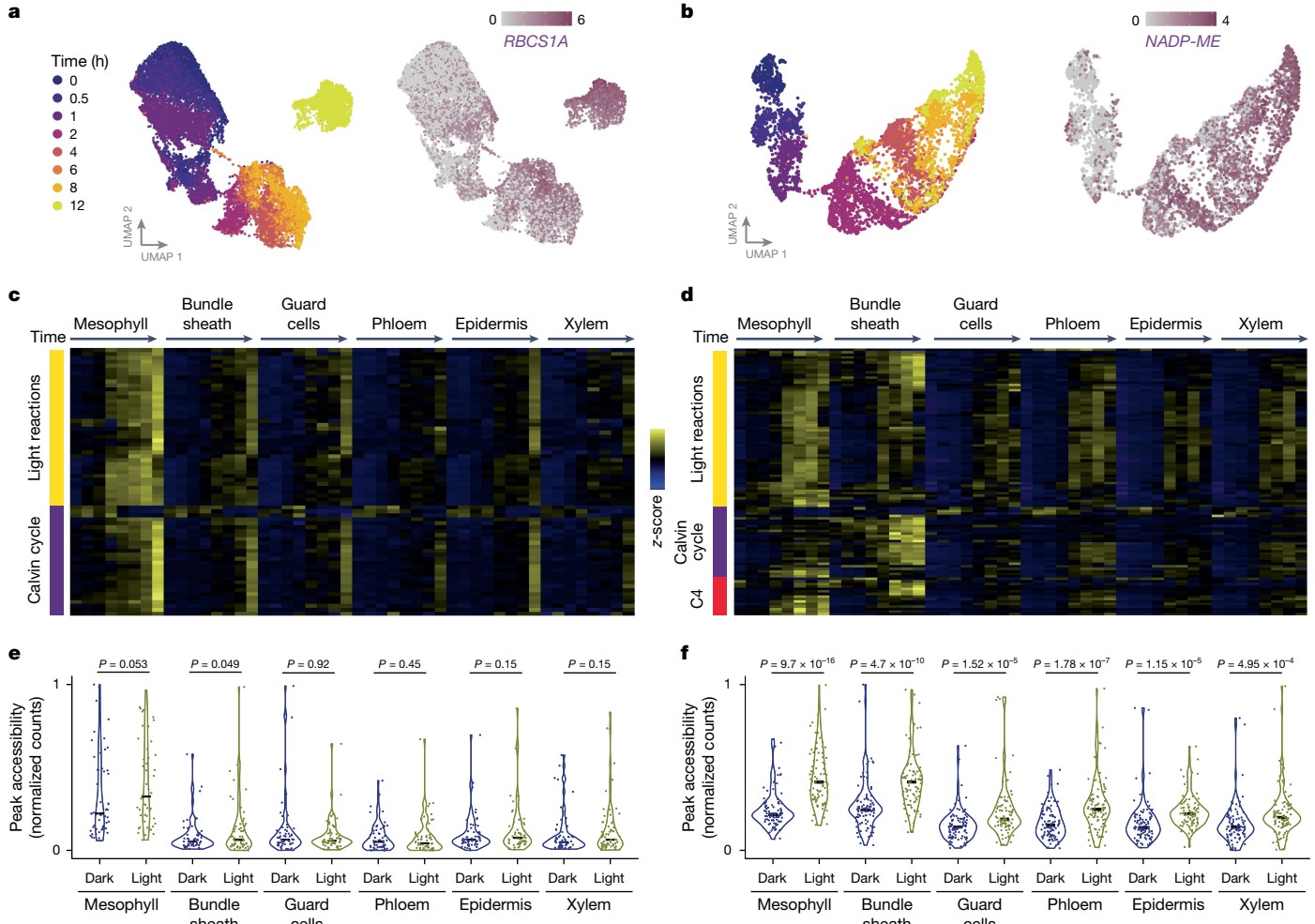

**Fig. 3 | Light changes cell-type-specific transcript abundance and chromatin accessibility. a**, Sub-clustering of rice mesophyll nuclei undergoing de-etiolation. The transcript abundance of *RBCS1A* is shown on the right. **b**, Sub-clustering of sorghum bundle-sheath nuclei undergoing de-etiolation. The transcript abundance of *NADP-ME* is shown on the right. **c,d**, Heat map of photosynthesis gene expression in different cell types of rice (**c**) and sorghum (**d**) during the first 12 h of light. Genes encoding proteins involved in $C_4$ photosynthesis, the Calvin–Benson–Bassham cycle and light reactions are shown in red, purple and yellow, respectively. **e,f**, Differences in the chromatin accessibility of photosynthesis genes in different cell types of rice (**e**) and sorghum (**f**) measured at 0 h (dark) and 12 h (light). Welch's *t*-test indicated.

this included genes involved in hormone signalling and biosynthesis, including the gibberellic acid, ethylene and auxin pathways, as well as genes that encode sugar and water transporters.

Next, we investigated how conserved cell-type-specific patterns of gene expression were across species (Supplementary Table 5). Although most cell types showed conserved patterns of expression between rice and sorghum, this was not the case for the bundle sheath (Fig. 2c and Extended Data Fig. 5). In fact, transcripts from only 31 orthologues (out of 229 rice bundle-sheath markers), including genes involved in sulfur metabolism and transport, were specific to the bundle sheath of both species (Fig. 2c and Supplementary Table 5). The $C_4$ bundle sheath of sorghum had also obtained patterns of gene expression from other cell types (Fig. 2c). Indeed, the bundle-sheath cells of sorghum were transcriptionally more similar to the mesophyll and guard cells of rice, whereas the bundle sheath of rice was most similar to the phloem of sorghum (Fig. 2c and Extended Data Fig. 5). Similarities between sorghum bundle sheath and rice mesophyll or guard cells were driven mainly by changes in the expression of genes involved in the Calvin–Benson–Bassham cycle and starch metabolism (Supplementary Table 5).

Because the differential expression of photosynthesis genes between bundle-sheath and mesophyll cells (hereafter, partitioning) is considered crucial for the evolution of $C_4$ photosynthesis[29] we examined this

phenomenon. A pairwise comparison of gene expression in response to light revealed that in each species, transcripts from more than 1,000 genes were partitioned between mesophyll and bundle-sheath cells, and this included 225 orthologues (Extended Data Fig. 5 and Supplementary Table 6). Of these, 126 were partitioned identically between the same cell types in both rice and sorghum (Fig. 2d). Of note, an additional 99 orthologues showed opposing patterns in the two species; that is, they were 'differentially' partitioned (Fig. 2d). Forty-three orthologues that had swapped from strong expression in the mesophyll of rice to strong expression in the bundle sheath of sorghum included genes encoding proteins of the Calvin–Benson–Bassham cycle, as well as organic acid and nitrogen metabolism. Fifty-six genes that swapped from strong expression in the bundle sheath of rice to the mesophyll of sorghum were associated with the transport of metabolites and solutes (Fig. 2d and Supplementary Table 6). We note that 12% of orthogroups had a more complex pattern of expression (Extended Data Fig. 5).

To investigate how conserved partitioning was between all cell types, we assessed the degree of cross-species overlap between each pair of cell types. This revealed that the mesophyll and bundle sheath had the smallest set of partitioned genes across species and the weakest statistical overlap (Extended Data Fig. 6 and Supplementary Table 7). In addition to mesophyll and bundle-sheath cells of rice showing the lowest

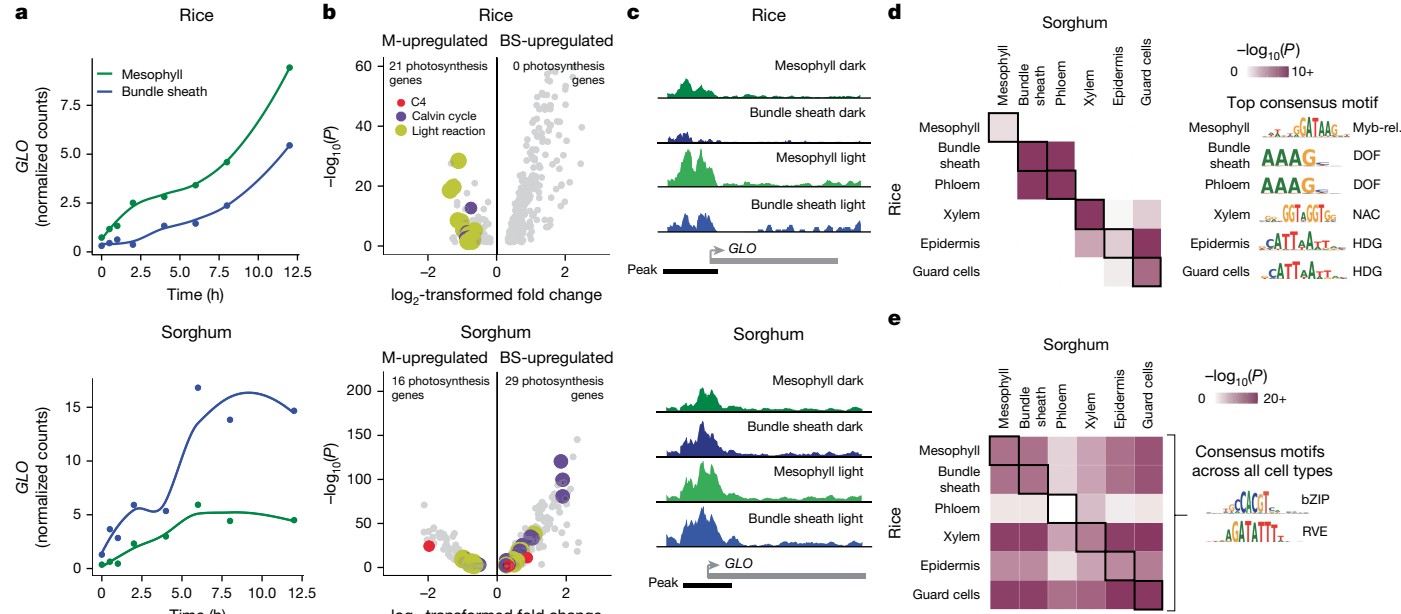

**Fig. 4 | Both cell identity and light drive the partitioning of photosynthesis genes between mesophyll and bundle-sheath cells. a**, Transcript abundance of the photorespiration gene *GLO* during de-etiolation in the mesophyll and bundle-sheath cells of rice and sorghum. Points indicate mean expression, line fit using locally estimated scatter plot smoothing. **b**, Volcano plots of genes significantly partitioned to either mesophyll (M) or bundle sheath (BS) under etiolated conditions (0-h time point, adjusted $P < 0.05$, likelihood-ratio test). Genes encoding proteins involved in $C_4$ photosynthesis, the Calvin–Benson–Bassham cycle and light reactions are shown in red, purple and yellow, respectively. **c**, Chromatin accessibility of *GLO* at 0 h (dark) and 12 h (light) in mesophyll and bundle-sheath nuclei. **d**, Overlap of *cis*-elements associated with accessible chromatin in each cell type of rice and sorghum (Fisher's exact test adjusted *P* indicated). The consensus motif for the most over-represented *cis*-element within each overlap is shown on the right (additional over-represented motifs in Supplementary Table 10). **e**, Overlap of *cis*-elements associated with accessible chromatin in each cell type in response to light in rice and sorghum (Fisher's exact test adjusted *P* indicated). The consensus motif for the most over-represented *cis*-element in all overlaps is shown on the right (additional over-represented motifs in Supplementary Table 12).

conservation in terms of transcript partitioning, it was also noticeable that a large proportion of genes that were partitioned between these cells had swapped cell types (Extended Data Fig. 6). This suggests that swapping of functions or 'identity' between other cell types is a rare event genome-wide but occurs relatively frequently between mesophyll and bundle sheath.

## Light regulation of the $C_4$ bundle sheath

Because light induces photomorphogenesis, we investigated how individual nuclei from each cell type responded to this stimulus. Rice mesophyll and sorghum bundle-sheath nuclei clustered by time of sampling, indicating that light was a dominant driver of transcriptional state (Fig. 3a,b). Canonical marker genes showed the expected induction. For example, *RBCS* and *NADP-ME* were activated by light in the mesophyll cells of rice and in the bundle-sheath cells of sorghum, respectively (Fig. 3a,b). Similarly, transcript abundances of light signalling transcription factors such as *ELONGATED HYPOCOTYL* (*HY5*) (ref. 31) and *PHYTOCHROME INTERACTING FACTORS 3–5* (*PIF3*, *PIF4* and *PIF5*) (refs. 32,33) were dynamic during de-etiolation (Extended Data Fig. 7). We detected global cell-type-specific differential gene-expression responses by fitting statistical models to pseudo-bulked transcriptional profiles. In rice, each of the six cell types showed a distinct and cell-type-specific response (Extended Data Fig. 7 and Supplementary Table 8). Apart from the bundle-sheath and epidermal cells in rice, hundreds of cell-type-specific light-responsive genes were detected (Extended Data Fig. 7). In both species, mesophyll- and bundle-sheath-specific genes were involved in photosynthesis and chloroplast-related functions, consistent with the rapid greening of shoots and conversion of etioplasts into chloroplasts (Supplementary Table 8). Bundle-sheath cells from rice and sorghum showed the greatest difference in their

response to light, with only 35 light-responsive bundle-sheath-specific genes detected in rice, but more than 1,000 genes induced by light in the bundle sheath of sorghum (Extended Data Fig. 7).

Light-induced partitioning of canonical photosynthesis genes between the mesophyll and bundle sheath was apparent (Fig. 3c,d and Supplementary Table 9). In rice, photosynthesis genes were most strongly induced in the mesophyll, although a similar but weaker response was also seen in the bundle sheath and other cell types (Fig. 3c). SEM confirmed that in the dark, etioplasts were present in vascular and epidermal cells, and after exposure to light, thylakoid-like membranes were evident (Extended Data Fig. 8). This supports the observation that photosynthesis can be weakly induced in these cell types. In sorghum, light strongly induced photosynthesis genes in both mesophyll and bundle-sheath cell types, and these included genes that are important for the light-dependent reactions of photosynthesis as well as the Calvin–Benson–Bassham and $C_4$ cycles (Fig. 3d). Agreeing with these data, chromatin of photosynthesis genes was more accessible in mesophyll cells than in other cell types ($P = 9.7 \times 10^{-16}$, Welch's *t*-test) (Fig. 3e,f), although the difference in accessibility in response to light was only marginally significant in the rice mesophyll ($P = 0.053$). By contrast, in sorghum, the accessibility of photosynthesis genes increased in response to light in both mesophyll and bundle-sheath cells ($P < 4.69 \times 10^{-10}$) (Fig. 3f). These data indicate that a pervasive gain of light regulation by photosynthesis genes in the bundle sheath of sorghum is likely to be facilitated by increased chromatin accessibility.

## Cell identity conditions gene expression

In both rice and sorghum, differences in the expression of photosynthesis genes between mesophyll and bundle-sheath cells increased

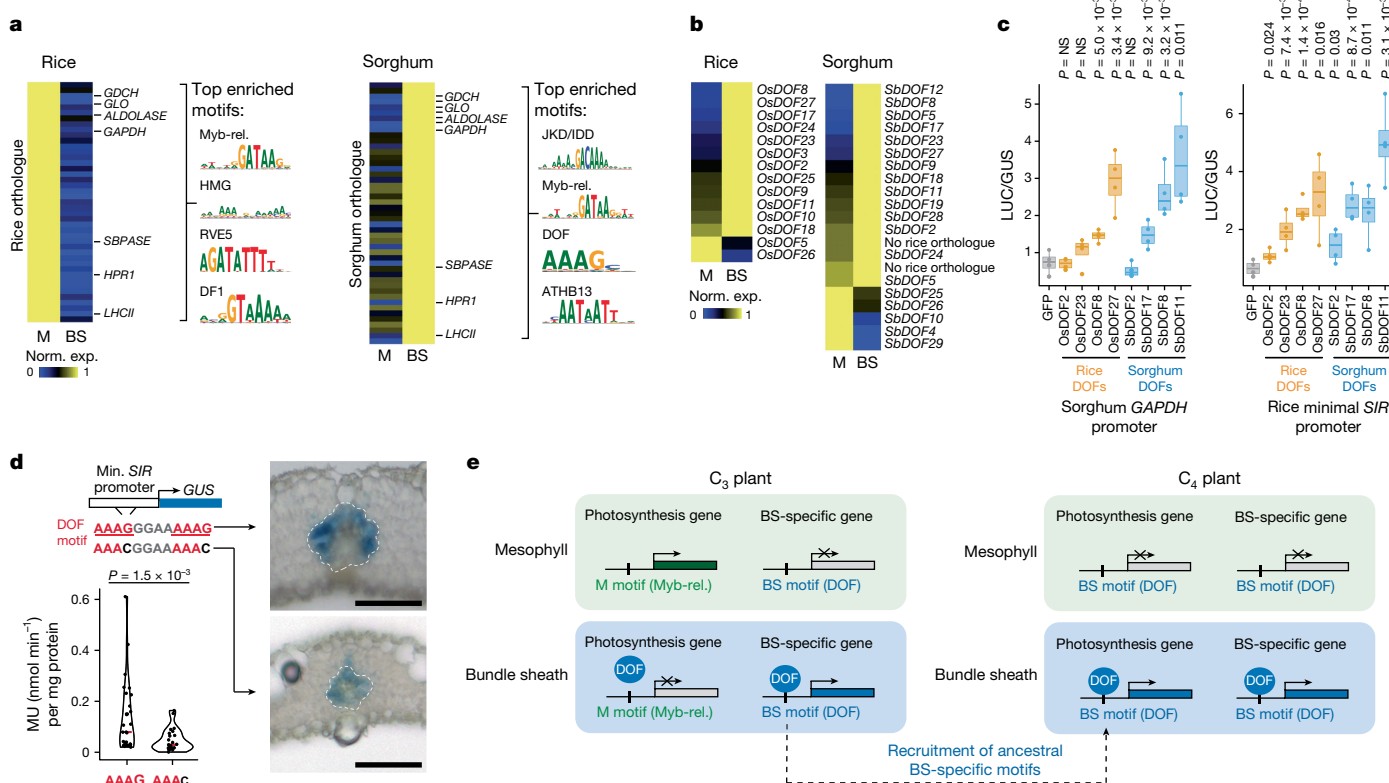

**Fig. 5 | A cell-type-specific cistrome in C₃ rice and C₄ sorghum drives the partitioning of photosynthesis between mesophyll and bundle-sheath cells.**
**a**, Gene-expression heat maps (left) of differentially partitioned orthologues in rice and sorghum, and the four most enriched *cis*-elements (right) in the accessible chromatin of corresponding genes. Additional over-represented motifs are shown in Supplementary Table 13. **b**, DOF transcription factor expression in mesophyll and bundle sheath in each species. Sorghum gene names based on orthology with rice genes. **c**, Transactivation of sorghum *GAPDH* promoter and rice minimal *SIR* promoter by DOF transcription factors from rice (orange) and sorghum (blue) (one-sided Welch's *t*-test *P* indicated; *n* = 4 biological replicates; boxes indicate 25th, median and 75th quartiles; whiskers

extend to the outermost value within 1.5× interquartile range; assay repeated three times independently with similar results). **d**, Activity of GUS reporter driven by minimal *SIR* promoter (containing two DOF motifs) in transgenic rice determined using the fluorometric 4-methylumbelliferyl-β-D-glucuronide assay (left). DOF motifs were mutated to replace G in AAAG with C (one-sided Welch's *t*-test *P* indicated; *n* = 29 independent transformants for minimal SIR promoter; *n* = 23 independent transformants for mutated DOF motifs). Representative cross-sections of GUS-stained transgenic leaves are shown on the right. Bundle-sheath cells are outlined with a dotted line. Scale bars, 50 μm. **e**, By acquiring DOF *cis*-elements, C₄ genes co-opt and amplify the ancestral bundle-sheath cell-identity network that is common between both species.

over time (Fig. 3c,d). This is exemplified by the *GLYCOLATE OXIDASE* (*GLO*) and *RBCS* genes, the transcripts of which showed greater partitioning to mesophyll cells of rice and bundle-sheath cells of sorghum in response to light (Fig. 4a and Extended Data Fig. 9). After 12 h of light, 72 photosynthesis genes in rice and 77 in sorghum were partitioned between mesophyll and bundle-sheath cells (Extended Data Fig. 9). However, for some photosynthesis genes, differences in expression between cells were already evident in the dark, suggesting that cell identity conditions light responses. Specifically, in the dark, 29% and 58% of photosynthesis transcripts in rice and sorghum, respectively, were significantly partitioned between mesophyll and bundle-sheath cells (Fig. 4b). This finding is consistent with the observation that promoters of photosynthesis genes contained regions of open chromatin in the etiolated state (Fig. 3e,f). In fact, in the etiolated state, many photosynthesis genes showed differences in chromatin accessibility between cell types, as exemplified by *GLO* and *RBCS* (Fig. 4c and Extended Data Fig. 9). And, in many instances, light exposure then increased chromatin accessibility, suggesting that light signalling enhances but does not establish chromatin accessibility within the promoters of photosynthesis genes (Extended Data Fig. 9). We conclude that intrinsic differences in cell identity contribute to the partitioning of photosynthesis gene expression between cells in both C₃ rice and C₄ sorghum, and that differential partitioning is not driven exclusively by light signalling.

## C₄ co-opts an ancestral *cis*-code

*Cis*-elements have a key role in driving the patterning of gene expression[34,35]. Therefore, we next searched for *cis*-elements that underlie the observed cell-identity- and light-dependent patterns of gene expression. When regions of open chromatin specific to each cell type were assessed for over-represented transcription-factor-binding sites, we found dozens of enriched *cis*-regulatory elements for each cell type (Supplementary Table 10). We compared the 25 most significantly enriched *cis*-regulatory motifs for each cell type across species. This identified a significant overlap of enriched motifs in the same cell types of both rice and sorghum (Fig. 4d and Supplementary Table 10). Thus, both species share a conserved cell-type-specific *cis*-regulatory code. For example, motifs bound by the myeloblastosis (Myb)-related, NAM, ATAF1, ATAF2 and CUC2 (NAC) transcription factors defined accessible chromatin regions in mesophyll nuclei from both rice and sorghum, whereas the DOF motif was enriched in bundle-sheath- and phloem-specific peaks of both species (Fig. 4d and Extended Data Fig. 10). We also detected DOF motif enrichment in promoter regions of homologues of rice and sorghum bundle-sheath partitioned genes in several other C₃ Poaceae species, including *Chasmanthium laxum*, *Hordeum vulgare* and *Brachypodium distachyon* (Extended Data Fig. 10 and Supplementary Table 11). By contrast, when we examined motifs in chromatin that were differentially accessible in response to

light, we found that the same motifs were enriched regardless of cell type. These comprised the light-responsive circadian-clock-related basic leucine zipper (bZIP) and CIRCADIAN CLOCK ASSOCIATED 1 (CCA1) motifs (Fig. 4e, Extended Data Fig. 11 and Supplementary Table 12). These findings suggest that cell-type-specific patterning of gene expression is defined by cell-identity *cis*-elements, whereas light-responsive gene expression is regulated by similar *cis*-elements shared by all cell types.

Next, we investigated whether the *cis*-code associated with each cell type regulates genes that are differentially partitioned between rice and sorghum. To this end, we examined genes that were preferentially expressed in the rice mesophyll, but whose orthologues were partitioned to the sorghum bundle sheath (Fig. 5a). Among the 40 orthologues in this category were the Calvin–Benson–Bassham cycle genes *FRUCTOSE BISPHOSPHATE ALDOLASE* and *GLYCERALDE-HYDE 3-PHOSPHATE DEHYDROGENASE* (*GAPDH*), photorespiration genes such as *GLO* and light reaction genes including the *LHCII* subunit (Fig. 5a and Supplementary Table 13). Notably, among these differentially partitioned genes, we found that associated chromatin was enriched in cell-type-specific Myb-related, high-mobility group (HMG), REVEILLE 5 (RVE5) and DF1 binding sites in mesophyll-specific genes in rice (Fig. 5a), but that it was enriched in cell-type-specific DOF and JACKDAW (JKD) or indeterminate domain (IDD) binding sites in bundle-sheath-specific orthologues in sorghum (Fig. 5a, Extended Data Fig. 12 and Supplementary Table 13). This indicates that these orthologues swapped their partitioning from mesophyll to bundle sheath by changing identity-associated *cis*-regulatory motifs. Specifically, our data indicate that DOF motifs were acquired by genes that swap expression from the mesophyll of $C_3$ rice to the bundle sheath of $C_4$ sorghum.

The consensus sequence common to all DOF-binding motifs is AAAG (ref. 36). Thus, we next examined the frequency of this core binding motif in open chromatin within 1,500 nucleotides of the transcription start site of differentially partitioned genes. Among these genes, there were more DOF-binding sites in accessible chromatin in sorghum compared with orthologues in rice (binomial $P = 5.5 \times 10^{-3}$; Extended Data Fig. 12). By contrast, we did not find such enrichment among consistently partitioned genes ($P = 0.48$; Extended Data Fig. 12). These trends were also evident in canonical photosynthesis genes expressed in the sorghum bundle sheath. For example, the accessible chromatin of sorghum *GAPDH* contained more than twice as many DOF motifs as the rice orthologue did, and a similar enrichment of DOF motifs was seen in the accessible chromatin of sorghum $C_4$ photosynthesis genes such as *NADP-ME* (Extended Data Fig. 12).

Furthermore, genes encoding DOF family transcription factors were typically more strongly expressed in the bundle-sheath cells of both rice and sorghum (Fig. 5b), suggesting that the cell-type patterning of these transcription factors has not changed during the transition from $C_3$ to $C_4$. To test whether DOF transcription factors are sufficient to regulate the expression of bundle-sheath-specific genes, we performed effector assays in rice protoplasts. Rice OsDOF8 and OsDOF27, as well as sorghum SbDOF17, SbDOF8 and SbDOF11, activated the expression of a *LUCIFERASE* reporter gene from the sorghum *GAPDH* promoter, whereas OsDOF2, OsDOF23 and SbDOF2 did not (Fig. 5c). Similarly, several rice and sorghum DOF transcription factors were able to activate expression from the sorghum *NADP-ME* promoter (Extended Data Fig. 12), as well as from the minimal Os*SIR* promoter, which drives bundle-sheath expression in rice[37] (Fig. 5c). Mutating two DOF-binding sites present in this minimal Os*SIR* promoter reduced GUS activity in stably transformed rice plants 2.8-fold (Fig. 5d). These data indicate that specific members of the DOF family are sufficient, and their cognate motifs are necessary for strong expression in the rice bundle sheath.

From our analysis we propose a model that explains the rewiring of cell-type-specific regulation of photosynthesis genes in $C_4$ leaves (Fig. 5e). The model suggests that (i) the same mesophyll- and bundle-sheath-specific *cis*-elements are active in rice and sorghum; (ii) patterning of transcription factors between the two species is relatively stable; and (iii) photosynthesis genes expressed in the sorghum bundle sheath have acquired DOF *cis*-elements associated with bundle-sheath cells in rice to amplify expression in this cell type.

## Discussion

Our findings indicate that the expression of $C_4$ genes in bundle-sheath cells is underpinned by the integration of two characteristics found in the ancestral $C_3$ state: first, a conserved cell-type-specific transcription factor network; and second, acquisition of ancestral cell identity *cis*-elements. In both $C_3$ rice and $C_4$ sorghum, DOF transcription factors were preferential to the bundle sheath, a finding strongly supporting the stability of the *trans*-network across $C_3$ and $C_4$ species. The acquisition of DOF-binding sites in the promoters of $C_4$ genes then allowed these genes to decode the pre-existing patterning of transcription factors to amplify expression in the bundle sheath. Together, this allowed $C_4$ evolution through the broadening of cell-identity networks.

It is noteworthy that DOF motifs were also enriched in genes expressed in the bundle sheath of $C_4$ maize[38], and transient assays led to the proposal that DOF transcription factors control bundle-sheath-specific genes such as maize *NADP-ME*[18]. Thus, a DOF-mediated mechanism for the regulation of bundle-sheath gene expression seems to be present in several $C_4$ species. Moreover, DOF motifs were not only enriched in bundle-sheath-specific genes from $C_3$ rice, but also in orthologues from *C. laxum*, a $C_3$ species from the PACMAD clade otherwise made up of $C_4$ grasses[39,40], as well as *H. vulgare* and *B. distachyon*, two $C_3$ grasses from the Poaceae[39]. Together, these results support a model in which DOF regulation of bundle-sheath expression is ancestral and has been co-opted during $C_4$ evolution to strengthen the expression of $C_4$ genes in this cell type.

Compared with previous analyses, the model we propose supports a distinct evolutionary trajectory leading to the $C_4$ state. For example, in $C_4$ dicotyledons, bundle-sheath- and mesophyll-specific gene expression has been reported to be mediated by existing *cis*-elements allowing changes in the *trans*-network to be decoded[14–16]. In contrast to previous work with monocotyledons, we were able to provide insight into *cis*-elements associated with bundle-sheath identity in a $C_3$ as well as a $C_4$ grass, and this allowed distinct insight into the evolutionary rewiring of cell-specific gene expression. Because we took an unbiased genome-wide approach, this reconfiguration in *cis* is likely to constitute one of the more notable changes in *cis*-code during $C_4$ evolution. Combined with previous analyses, which indicated that the acquisition of pre-existing *cis*-elements associated with light-responsive elements allows enhanced $C_4$ gene expression in $C_4$ compared with $C_3$ leaves of dicotyledons[25], our data argue for evolution repeatedly making use of an existing *trans*-code through pervasive rewiring in *cis*.

These findings have key implications for ongoing efforts to engineer $C_3$ plants with $C_4$ photosynthetic characteristics[41]. For example, considerable efforts have been made to install the biochemistry of $C_4$ photosynthesis in rice[3,4,42]. However, a major obstacle has so far been the robust expression of $C_4$ acid decarboxylase genes in the rice bundle sheath[3,41]. The data reported here indicate that rewiring such genes to recognize an ancient bundle-sheath-identity network based on DOF transcription factors could be a useful way forward.

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

## Methods

### Plant growth

For the de-etiolation time course, seeds of *Oryza sativa* spp. *japonica* cultivar Kitaake and *Sorghum bicolor* BTx623 were incubated in sterile water for two days and one day, respectively, at 29 °C in the dark. Germinated seedlings were transferred in a dark room equipped with green light to a 1:1 mixture of topsoil and sand supplemented with fertilizer granules and grown for five days in the dark by wrapping the tray and lid several times with aluminium foil. Plants were placed in a controlled environment room with 60% humidity and temperatures of 28 °C and 20 °C during the day and night, respectively. Plants were exposed to light at the beginning of a photoperiod of 12 h light and 12 h dark and shoots were harvested at different time points during de-etiolation by flash-freezing tissue in liquid nitrogen. For the 0-h time point, seedlings were harvested in a dark room equipped with green light and flash-frozen immediately.

For microscopy analysis and enrichment of bundle-sheath nuclei using fluorescence-activated nuclei sorting, *O. sativa* spp. *japonica* cultivar Kitaake single-copy homozygous T2 seeds were de-husked and sterilized in 10% (v/v) bleach for 30 min. After washing several times with sterile water, seeds were incubated for two days in sterile water at 29 °C in the dark. Germinated seedlings were transferred to half-strength Murashige and Skoog medium with 0.8% agar in Magentas and grown for five days in the light in a growth chamber at temperatures of 28 °C and 20 °C during the day and night, respectively, and a photoperiod of 12 h light and 12 h dark.

### Construct design and cloning

To generate constructs for the rice bundle-sheath marker line, the coding sequence for mTurquoise2 was obtained from a previous report[43], and the promoter sequence from *Zoysia japonica PHOSPHOENOLPYRUVATE CARBOXYKINASE* in combination with the dTALE STAP4 system was obtained from a previous report[44]. The coding sequence of *Arabidopsis thaliana* H2B (At5g22880) was used as an N-terminal signal for targeting mTurquoise2 to the nucleus. All sequences were domesticated for Golden Gate cloning[45,46]. Level 1 and Level 2 constructs were assembled using the Golden Gate cloning strategy to create a binary vector for the expression of STAP4-mTurquoise2-H2B driven by PCK-dTALE.

For the transactivation assay in rice protoplasts, transcription factor coding sequences were amplified using rice leaf cDNA or synthesized using GeneArt after domesticating the sequences for Golden Gate cloning[41,42] (*OsDOF2*, LOC_Os01g15900, *OsDOF8*, LOC_Os02g45200, *OsDOF23*, LOC_Os07g32510, *OsDOF27*, LOC_Os10g26620, *SbDOF2*, Sobic.003G121400, *SbDOF8*, Sobic.004G284400, *SbDOF11*, Sobic.001G489900 and *SbDOF17*, Sobic.006G182300). The coding sequences were assembled into a Level 1 module with a *Zea mays UBI* promoter and *Tnos* terminator module as described previously[37]. For the minimal *SIR* promoter, nucleotides −980 to −829, as well as the endogenous core promoter (nucleotides −250 to +42), were fused with the *LUCIFERASE* reporter to measure transcription activity[37].

To generate GUS reporter rice lines, the minimal *SIR* promoter was assembled into a Level 1 module with the coding sequence for *kzGUS* (an intronless version of the GUS reporter gene) and the *Tnos* terminator as described previously[37]. The DOF motifs in the minimal SIR promoter were mutated using PCR amplification.

### Rice transformation

*Oryza sativa* spp. *japonica* cultivar Kitaake was transformed using *Agrobacterium tumefaciens* as described previously[47], with several modifications. Seeds were de-husked and sterilized with 10% (v/v) bleach for 15 min before placing them on nutrient broth (NB) callus induction medium containing 2 mg l⁻¹ 2,4-dichlorophenoxyacetic acid for four weeks at 28 °C in the dark. Growing calli were co-incubated with *A. tumefaciens* strain LBA4404 carrying the expression plasmid of interest in NB inoculation medium containing 40 μg ml⁻¹ acetosyringone for three days at 22 °C in the dark. Calli were transferred to NB recovery medium containing 300 mg⁻¹ timentin for one week at 28 °C in the dark. They were then transferred to NB selection medium containing 35 mg l⁻¹ hygromycin B for four weeks at 28 °C in the dark. Proliferating calli were subsequently transferred to NB regeneration medium containing 100 mg l⁻¹ myo-inositol, 2 mg l⁻¹ kinetin, 0.2 mg l⁻¹ 1-naphthaleneacetic acid and 0.8 mg l⁻¹ 6-benzylaminopurine for four weeks at 28 °C in the light. Plantlets were transferred to NB rooting medium containing 0.1 mg l⁻¹ 1-naphthaleneacetic acid and incubated in Magenta pots for two weeks at 28 °C in the light. Finally, plants were transferred to a 1:1 mixture of topsoil and sand and grown in a controlled environment room with 60% humidity, temperatures of 28 °C and 20 °C during the day and night, respectively, and a photoperiod of 12 h light and 12 h dark.

### Transactivation assay

Rice leaf protoplast isolation was performed as described previously[37,48]. Protoplasts were transformed using Golden Gate Level 1 modules designed for constitutive expression of transcription factors, alongside the *LUC* reporter and the *ZmUBIpro::GUS-Tnos* transformation control, which were prepared with the ZymoPURE II Plasmid Midiprep Kit. The transformation mixture contained 2 μg of control plasmids, 5 μg of reporter plasmids and 5 μg of transcription factor plasmids, which were transformed into 180 μl of protoplasts. After incubating protoplasts for 20 h in the light, proteins were extracted using passive lysis buffer (Promega), and GUS activity was measured with 20 μl of the protein extract. A fluorometric MUG (4-methylumbelliferyl-β-ᴅ-glucuronide) assay was used for quantifying GUS activity[49] in a reaction mixture of 200 μl containing 50 mM phosphate buffer (pH 7.0), 10 mM EDTA-Na₂, 0.1% (v/v) Triton X-100, 0.1% (w/v) *N*-lauroylsarcosine sodium, 10 mM DTT and 2 mM MUG. The assay was performed at 37 °C, and 4-methylumbelliferone (4-MU) fluorescence was recorded every 2 min for 20 cycles at 360 nm excitation and 450 nm emission using a CLARIOstar plate reader. In addition, LUC activity was determined using 20 μl of protein sample and 100 μl of LUC assay reagent from Promega. Transcription activity was quantified as LUC luminescence relative to the rate of MU accumulation per second.

### GUS staining

GUS staining was performed as described previously[49], with minor modifications. Leaf tissue was fixed in 90% (v/v) acetone for 12 h at 4 °C. After washing with 100 mM phosphate buffer (pH 7.0), samples were transferred into 1 mg ml⁻¹ 5-bromo-4-chloro-3-indolyl glucuronide (X-Gluc) GUS staining solution and vacuum was applied five times for 2 min each. The samples were incubated at 37 °C for 48 h. To clear chlorophyll, samples were incubated in 90% (v/v) ethanol at room temperature. Cross-sections were prepared with a razor blade and images were taken with an Olympus BX41 light microscope.

To quantify GUS activity, a fluorometric MUG assay was used[49] as described above, using 200 mg of mature leaf tissue. A standard curve of ten 4-MU standards was used to determine the 4-MU concentration in each sample.

### Confocal microscopy

To test the bundle-sheath-specific expression of mTurquoise2-H2B, recently expanded leaf 3 of seven-day-old seedlings was prepared for confocal microscopy by scraping the adaxial side of the leaf blade two to three times with a sharp razor blade, transferring to water to avoid drying out and then mounting on a microscope slide with the scraped surface facing upwards. Confocal imaging was performed on a Leica TCS SP8 X using a 10× air objective (HC PL APO CS2 10×0.4 Dry) with optical zoom, and hybrid detectors for fluorescent protein and chlorophyll autofluorescence detection. The following excitation (Ex) and emission (Em) wavelengths were used for imaging: mTurquoise2 (Ex = 442, Em = 471–481), chlorophyll autofluorescence (Ex = 488, Em = 672–692).

## SEM

For the de-etiolation experiment of rice and sorghum, samples from four to six individual seedlings for each time point (0 h, 6 h, 12 h and 48 h) were collected for electron microscopy. Leaf segments (around 2 mm²) were excised with a razor blade and immediately fixed in 2% (v/v) glutaraldehyde and 2% (w/v) formaldehyde in 0.05–0.1 M sodium cacodylate (NaCac) buffer (pH 7.4) containing 2 mM calcium chloride. Samples were vacuum infiltrated overnight, washed five times in 0.05–0.1 M NaCac buffer and post-fixed in 1% (v/v) aqueous osmium tetroxide, 1.5% (w/v) potassium ferricyanide in 0.05 M NaCac buffer for three days at 4 °C. After osmication, samples were washed five times in deionized water and post-fixed in 0.1% (w/v) thiocarbohydrazide for 20 min at room temperature in the dark. Samples were then washed five times in deionized water and osmicated for a second time for 1 h in 2% (v/v) aqueous osmium tetroxide at room temperature. Samples were washed five times in deionized water and subsequently stained in 2% (w/v) uranyl acetate in 0.05 M maleate buffer (pH 5.5) for three days at 4 °C and washed five times afterwards in deionized water. Samples were then dehydrated in an ethanol series, and transferred to acetone and then to acetonitrile. Leaf samples were embedded in Quetol 651 resin mix (TAAB Laboratories Equipment) and cured at 60 °C for two days. Ultra-thin sections of embedded leaf samples were prepared and placed on Melinex (TAAB Laboratories Equipment) plastic coverslips mounted on aluminium SEM stubs using conductive carbon tabs (TAAB Laboratories Equipment), sputter-coated with a thin layer of carbon (around 30 nm) to avoid charging, and imaged in a Verios 460 scanning electron microscope at a 4 keV accelerating voltage and 0.2 nA probe current using the concentric backscatter detector in field-free (low-magnification) or immersion (high-magnification) mode (working distance 3.5–4 mm, dwell time 3 μs, 1,536 × 1,024 pixel resolution). For overserving plastid ultrastructure, SEM stitched maps were acquired at 10,000× magnification using the FEI MAPS automated acquisition software. Greyscale contrast of the images was inverted to allow easier visualization.

## Enrichment of bundle-sheath nuclei using fluorescence-activated cell sorting

To purify the nuclei population from whole leaves, recently expanded leaves 3 from five seven-day-old wild-type rice seedlings were chopped on ice in nuclei buffer (10 mM Tris-HCl, pH 7.4, 10 mM NaCl, 3 mM MgCl$_2$, 0.5 mM spermidine, 0.2 mM spermine, 0.01% Triton X, 1× Roche complete protease inhibitors, 1% BSA and Protector RNase inhibitor) with a sharp razor blade. The suspension was filtered through a 70-mm filter and subsequently through a 35-mm filter. Nuclei were stained with Hoechst and purified by fluorescence-activated cell sorting (FACS) on an AriaIII instrument, using a 70-mm nozzle. Nuclei were collected in an Eppendorf tube containing BSA and Protector RNase inhibitor. Using the same approach, nuclei from the bundle-sheath marker line expressing mTurquoise2-H2B were isolated. Nuclei were sorted on the basis of the mTurquoise2 fluorescent signal. Nuclei were collected in minimal nuclei buffer (10 mM Tris-HCl, pH 7.4, 10 mM NaCl, 3 mM MgCl$_2$, RNase inhibitor and 0.05% BSA). After collection, nuclei were spun down in a swinging bucket centrifuge at 405$g$ for 5 min, with reduced acceleration and deceleration. Nuclei were resuspended in minimal nuclei buffer and mixed with the unspun whole leaf nuclei population to achieve a proportion of approximately 25% mTurquoise2-positive nuclei. The bundle-sheath enriched nuclei population was sequenced using the 10X Genomics Gene Expression platform with v.3.1 chemistry, and sequenced on the Illumina NovaSeq 6000 with 150-bp paired-end chemistry.

## Chlorophyll quantification

Seedlings were harvested at specified time points during de-etiolation and immediately flash-frozen in liquid nitrogen. Frozen tissue was ground into fine powder and the weight was measured before suspending the tissue in 1 ml of 80% (v/v) acetone. After vortexing, the tissue was incubated on ice for 15 min with occasional mixing of the suspension. The tissue was spun down at 15,700$g$ at 4 °C and the supernatant was removed. The extraction was repeated, and supernatants were pooled before measuring the absorbance at 663.6 nm and 646.6 nm in a spectrophotometer. The total chlorophyll content was determined as described previously[50].

## Nuclei extraction and single-nucleus RNA-seq (10X RNA-seq)

Frozen tissue from each time point (one biological replicate per time point, eight time points) was crushed using a bead bashing approach, and nuclei were released from homogenate by resuspending in nuclei buffer (10 mM Tris-HCl, pH 7.4, 10 mM NaCl and 3 mM MgCl$_2$). The resulting suspension was passed through a 30-μm filter. To enrich the filtered solution for nuclei, an Optiprep (Sigma) gradient was used. Enriched nuclei were then stained with Hoechst, before being FACS purified (BD Influx Software v.1.2.0.142). Purified nuclei were run on the 10X Gene Expression platform with v.3.0 chemistry, and sequenced on the Illumina NovaSeq 6000 with 150-bp paired-end chemistry. Single-cell libraries were made following the manufacturers protocol. Libraries were sequenced to an average saturation of 63% (14% s.d.) and aligned either to the rice (*O. sativa*, subspecies *Nipponbare*; MSU annotation)[51] or sorghum (*S. bicolor* v.3.0.1; JGI annotation)[52] genome. Chloroplast and mitochondrial reads were removed. For each time point, an average of 12,524 nuclei were sequenced (6,405 s.d.), with an average median unique molecular identifier (UMI) of 1,152 (420 s.d.) across both species. Doublets were removed using doubletFinder[53].

## Nuclei extraction and single-nucleus RNA-seq (sci-RNA-seq3)

Each individual frozen seedling (10–12 individual seedlings per time point) was crushed using a bead bashing approach in a 96-well plate, after which homogenate was resuspended in nuclei buffer. Resulting suspensions were passed through a 30-μm filter. Washed nuclei were then reverse-transcribed with a well-specific primer. After this step, remaining pool and split steps for sci-RNA-seq3 were followed as outlined previously[26]. We note the same approach was used to sequence the 48-h time point; however, a population of six plants was used instead of individual seedlings. Libraries were sequenced to an average saturation of 80% (5% s.d.), and sequenced on the Illumina NovaSeq 6000 with 150-bp paired-end chemistry. Reads were aligned to either the rice or the sorghum genome, as described above. Chloroplast and mitochondrial reads were removed. For 0–12-h time points, an average of 6,527 nuclei were sequenced (5,039 s.d.), with an average median UMI of 423 (41 s.d.) across both species. For the 48-h time point, 77,208 and 82,748 nuclei were sequenced with a median UMI of 757 and 740 for rice and sorghum, respectively.

## Nuclei extraction and single-nucleus RNA-seq (10X Multiome)

Fresh seedling tissue was collected after 0 or 12 h light treatment (two biological replicates per species, each with two to four technical replicates per time point; $n = 11$). Fresh tissue was chopped finely on ice in green room conditions in nuclei buffer. The resulting homogenate was filtered using a 30-μm filter. Nuclei were enriched using Optiprep gradient. No FACS was performed. Nuclei were run on the 10X Multiome platform with v.1.0 chemistry. Single-cell libraries were made following the manufacturer's protocol, and sequenced on the Illumina NovaSeq 6000 with 150-bp paired-end chemistry. Reads were aligned to either the rice or the sorghum genome, as described above. Chloroplast and mitochondrial reads were removed. For each sample, an average of 1,923 nuclei were sequenced (1,334 s.d.), with an average median UMI of 1,644 (646 s.d.) and median ATAC fragments 10,251 (7,001 s.d.) across both species.

## Nuclei clustering

Transcriptional atlases were generated separately for each species using Seurat[54]. Nuclei were first aggregated across various time points

(ranging from 0 to 48 h) and methods (10X and sci-RNA-seq3). The integrated dataset was subjected to clustering, using the top 2,000 variable features that were shared across all datasets. Each cluster contained nuclei sampled from all time points, indicating that clustering was driven predominantly by cell type rather than by time after exposure to light (Extended Data Fig. 2). Subsequent UMAP projections were constructed using the first 30 principal components. UMAP projections of mesophyll and bundle-sheath sub-clusters in rice and sorghum, respectively, were achieved using genes found to be significantly differentially expressed in response to light as variable features. To analyse the rice bundle-sheath-specific mTurquoise line, we integrated two treatment replicates into a unified dataset. For this dataset, we clustered using the first 30 principal components. Cluster-specific markers were identified using the FindMarkers() command (adjusted P value < 0.01). To determine the correspondence between the mTurquoise-positive cluster and clusters within the rice-RNA atlas, we compared the lists of cluster-specific markers (adjusted P value 0.01, specificity > 2) to those obtained from the rice atlas. For the 10X-multiome (RNA + ATAC) clustering we used Signac[55]. Biological and technical replicates for each species were integrated, and clustering was conducted using the first 50 principal components derived from expression data. After the initial peak calling using Cell Ranger (10X Genomics), peaks were subsequently re-called using MACS2 (ref. 56). Differentially accessible peaks between cell types were identified using the FindMarkers() command (adjusted P value < 0.05, per cent threshold > 0.3), before being associated with the nearest gene (±2,000 bp from transcription start site)

## Orthology analyses

We determined gene orthologues between rice and sorghum using OrthoFinder[57]. We constructed pan-transcriptome atlases by selecting expressed rice and sorghum genes that had cross-species orthologues. To construct the pan-transcriptome atlas, orthologue conversions were performed in a one-to-one manner, meaning that if multiple orthologues for a gene were found across species, only one was retained. We integrated these datasets with Seurat using the clustering approaches described above. To assign cell identities, we drew on cell-type labels that were previously assigned to each species separately and mapped them onto the pan-transcriptome clusters. To assess specific transcriptional differences in gene expression between the bundle-sheath clusters of sorghum and rice within this dataset, we used the FindMarkers() command (adjusted P value < 0.05). Sorghum DOF transcription factor orthologue names kept the same numerical identifier as their rice orthologues.

To examine the overlap of cell-type-specific gene-expression markers between the two species, we identified cell-type markers from our main transcriptional dataset using FindMarkers() (adjusted P value < 0.05, min.pct > 0.1). We note that some genes were found to be significant across multiple cell types. To assess the significance of the overlap between cell types across species, we converted genes to orthogroups and conducted a Fisher's exact test, with the total number of orthogroups in the dataset as the background. The proportion of conserved marker genes for each cell type across species ranged from 43% for mesophyll (184 out of 426 rice marker genes conserved in sorghum) to 13% for bundle sheath (31 out of 229 rice marker genes conserved in sorghum). We note that by relying on orthogroups, we included higher-order orthology relationships beyond a one-to-one manner.

Next, we assessed consistent and differential partitioning of gene-expression patterns among each cell-type pair (15 pairs total). To do this, we first calculated differentially expressed genes for each cell-type pair by pseudo-bulking transcriptomes of individual cell types across 0–12-h time points. Next, we identified partitioned expression patterns between cell types using an ANCOVA model implemented in DESeq2 (adjusted P value < 0.05). To perform cross-species comparisons of cell-type pairs, we first converted differentially expressed genes to their orthogroup. We then overlapped each cell-type pair across

species, using orthogroup membership, and evaluated the significance of these overlaps using the Fisher's exact test, with the total number of orthogroups as background. Finally, to distinguish whether a gene displayed consistent or differential partitioning in a particular cell type, we examined whether its fold change expression was higher or lower compared with its counterpart in the corresponding cell type of the other species.

## Differential expression and accessibility responses to light

We discovered cell-type-specific differentially expressed genes during the first 12 h of light by pseudo-bulking transcriptional profiles. To create pseudo-bulk profiles for each cell type, we first refined our nuclei clusters through re-clustering mesophyll, epidermal and vasculature cell classes separately, before selecting sub-clusters that most strongly expressed known cell-type marker genes. For each cell type, we calculated the first and second principal component of these bulked profiles and found differentially expressed genes through fitting linear models to each of these principal components, as well as those that responded linearly with time using DESeq2 (adjusted P < 0.05). We treated the assay with which the nuclei were sequenced (10X or sci-RNA-seq3) as a covariate. In this list of differentially expressed genes, we also included genes that were differentially expressed between time points 0 h and 12 h in a pairwise test (adjusted P < 0.05). Next, to uncover the different trends of gene expression among differentially expressed genes, we clustered genes using hierarchical clustering, choosing clustering cut-offs that resulted in 10 rice and 18 sorghum clusters that contained at least 10 genes. To visualize the expression of these clusters, we scaled the expression and fitted a non-linear model to capture the dominant expression trend. Accessible chromatin within canonical photosynthesis genes was found through pseudo-bulking accessible chromatin by cell type. Accessible peaks needed to be within 2,000 bp of the gene body. Only one peak per gene was retained for subsequent analyses, and extreme outliers were removed (around 5% of called peaks). To compare peak accessibility across species, reads per peak were re-normalized between 0 and 1. Significant differences in accessibility between cell types of this group of genes were assessed using a Student's t-test (one-sided).

## GO analyses

To identify GO terms associated with cell-type-specific genes and genes that swap expression patterns in rice and sorghum leaves, we performed singular enrichment analysis using the web-based tool AgriGO v.2.0 (ref. 58). *Oryza sativa* or *S. bicolor* gene identifiers were used for the input sample list, and the whole genome of the respective plant species was used as background.

## *Cis*-element analyses

We detected cell-type-specific accessible motifs within each cell type using the chromVAR function[59] implemented in Signac. In brief, this approach detected over-represented *cis*-regulatory elements within the JASPAR2020 plant taxon group[60] among peaks that are differentially accessible across cell-type clusters. GC enrichment and genomic backgrounds used for statistical tests were derived from BSGenome assembled genomes[61]. The same approach was also used to detect light-responsive *cis*-elements, using light- and dark-treated nuclei within each cell type. We overlapped enriched *cis*-regulatory elements identified across species by selecting the top 25 most significantly over-represented motifs (adjusted P < 0.05), before computing a Fisher's exact test using all computed motifs as background, and then clustered the resulting motifs using TOBIAS[62].

To find consistently and differentially partitioned orthologous genes within our multiome gene-expression dataset, we found mesophyll and bundle-sheath-specific genes in rice and sorghum, respectively, using the FindMarkers() command, with a P value threshold cut-off of 0.01 and an expression specificity above 1.25. To find over-represented motifs

within differentially partitioned genes, we correlated peak accessibility with gene expression using the LinkPeaks() command and kept only those peaks which were significantly associated with gene expression. We identified enriched *cis*-elements within these peaks using the FindMotifs() command; ranking by significance (adjusted $P < 0.05$). Because the resulting significance depends on the subset of the genome chosen as background, we iterated the FindMotifs() command over 100 permutations to rank motifs that were consistently reported as enriched. We then averaged each motif's respective rank across the 100 permutations to create a final ranked value (Supplementary Table 13).

To quantify the occurrence of DOF-binding sites, we extracted the genomic sequence of peaks that were proximal to the transcription start site (±1,500 bp). If a peak was proximal to two transcription start sites, it was assigned to the closer one. We then implemented Find Individual Motif Occurrences (FIMO) to quantify the number of DOF consensus sites within these chromatin regions ($P$ value threshold = 0.005). We chose the DOF2 (MA0020.1) motif as representative of the core DOF consensus sequence AAAG.

We implemented analysis of motif enrichment (AME) to detect DOF transcription factor motifs enriched within *C. laxum* (http://phytozome-next.jgi.doe.gov/info/Claxum_v1_1), *H. vulgare* (Hvulgare_r1)[63] or *B. distachyon* (Bdistachyon_314_v3.0)[64] homologues of genes consistently partitioned to the rice and sorghum bundle sheath. To identify homologues, the NCBI BLASTN tool v.2.15.0 was used by comparing coding sequences, and the top identified homologue for each gene was selected for *cis*-element enrichment analyses. We used 1,000 bp upstream of the transcription start site for each homologous gene and tested against reported plant motifs present within the JASPAR database.

## Reporting summary

Further information on research design is available in the Nature Portfolio Reporting Summary linked to this article.

## Data availability

Raw and processed data, including assembled atlases, have been deposited at the Gene Expression Omnibus and are publicly available at GSE248919. Raw data for chlorophyll measurement, transactivation assays and GUS quantification have been deposited at Mendeley (https://doi.org/10.17632/6xmsdg9xcr.1). Microscopy data reported in this paper will be shared by the J.M.H. upon request. Any additional information required to reanalyse the data reported in this paper is available from J.M.H. upon request.

## Code availability

The code for bioinformatic analyses presented in this manuscript is available on GitHub at https://github.com/joey1463/C3-C4.git.

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

**Acknowledgements** J.S. is an awardee of the Life Sciences Research Foundation (funded by Open Philanthropy) and a recipient of the American–Australian Association Fellowship (funded by Pratt Industries). L.H.L. is a recipient of the Herchel Smith Postdoctoral Research Fellowship and was supported by BBSRC grant BBP0031171 to J.M.H. L.H., N.W. and S.S. were supported by a Bill and Melinda Gates Foundation C₄ Rice grant awarded to the University of Oxford (2019-2024 INV-002970). T.B.S. was supported by a Swiss National Science Foundation (SNSF) Postdoc Mobility Fellowship (P500PB_203128) and an EMBO Long-Term Fellowship (ALTF 531-2019). J.R.E. is an Investigator of the Howard Hughes Medical Institute. We acknowledge the flow cytometry facility from the School of the Biological Sciences (University of Cambridge) for their support and assistance in this work. We thank K. H. Müller, G. E. Lindop and M. J. Drignon from the Cambridge Advanced Imaging Centre for the electron microscopy sample preparation and for support during image acquisition. The sequence data for *C. laxum* were produced by the Department of Energy's Joint Genome Institute.

**Author contributions** L.H.L., J.S., J.R.E. and J.M.H. designed the experimental plan. J.S and L.H.L. performed laboratory experiments and genomic analyses. L.H and N.W. performed transactivation assays and GUS expression analyses of stable rice lines. T.B.S. performed SEM imaging. R.M.D. and S.S. performed stable rice transformation. T.A.L. optimized nuclei isolation. J.R.N. performed Illumina sequencing. L.H.L., J.S., J.M.H. and J.R.E. wrote the manuscript, with input from all authors.

**Competing interests** The authors declare no competing interests.

**Additional information**
**Correspondence and requests for materials** should be addressed to Leonie H. Luginbuehl, Joseph R. Ecker or Julian M. Hibberd.

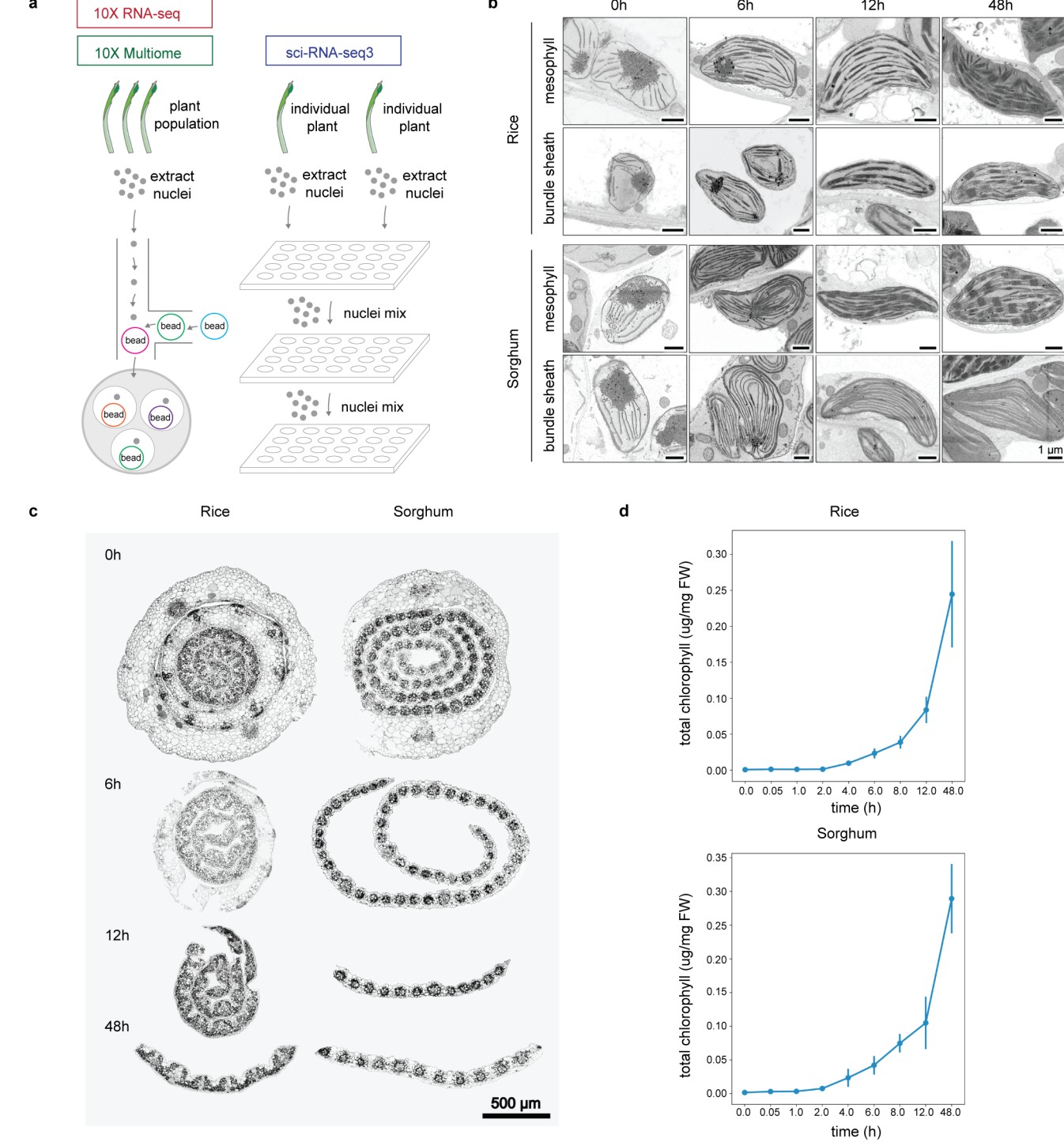

**Extended Data Fig. 1 | Single-nucleus sequencing of rice and sorghum shoots during de-etiolation. a**, Summary of 10X Genomics platform used for RNA and ATAC-sequencing of single nuclei extracted from a population of plants (adapted from Zheng et al.[65]), and sci-RNA-seq3 used for sequencing single nuclei from individual plants to provide increased biological replication. **b**, Scanning electron micrographs (SEMs) of etioplasts and chloroplasts of rice and sorghum mesophyll and bundle-sheath cells after 0 h, 6 h, 12 h and 48 h of light exposure (SEMs consistent across 3 biological replicates). **c**, SEMs of rice and sorghum leaf cross-sections showing leaf maturation from 0 h to 48 h after light exposure. (SEMs consistent across 3 biological replicates). **d**, Total chlorophyll (chlorophyll a + chlorophyll b) measured at different time points during de-etiolation in rice and sorghum. Each data point represents the mean of 3 biological replicates, +/− standard deviation from the mean. The experiment was repeated 3 times independently with similar results.

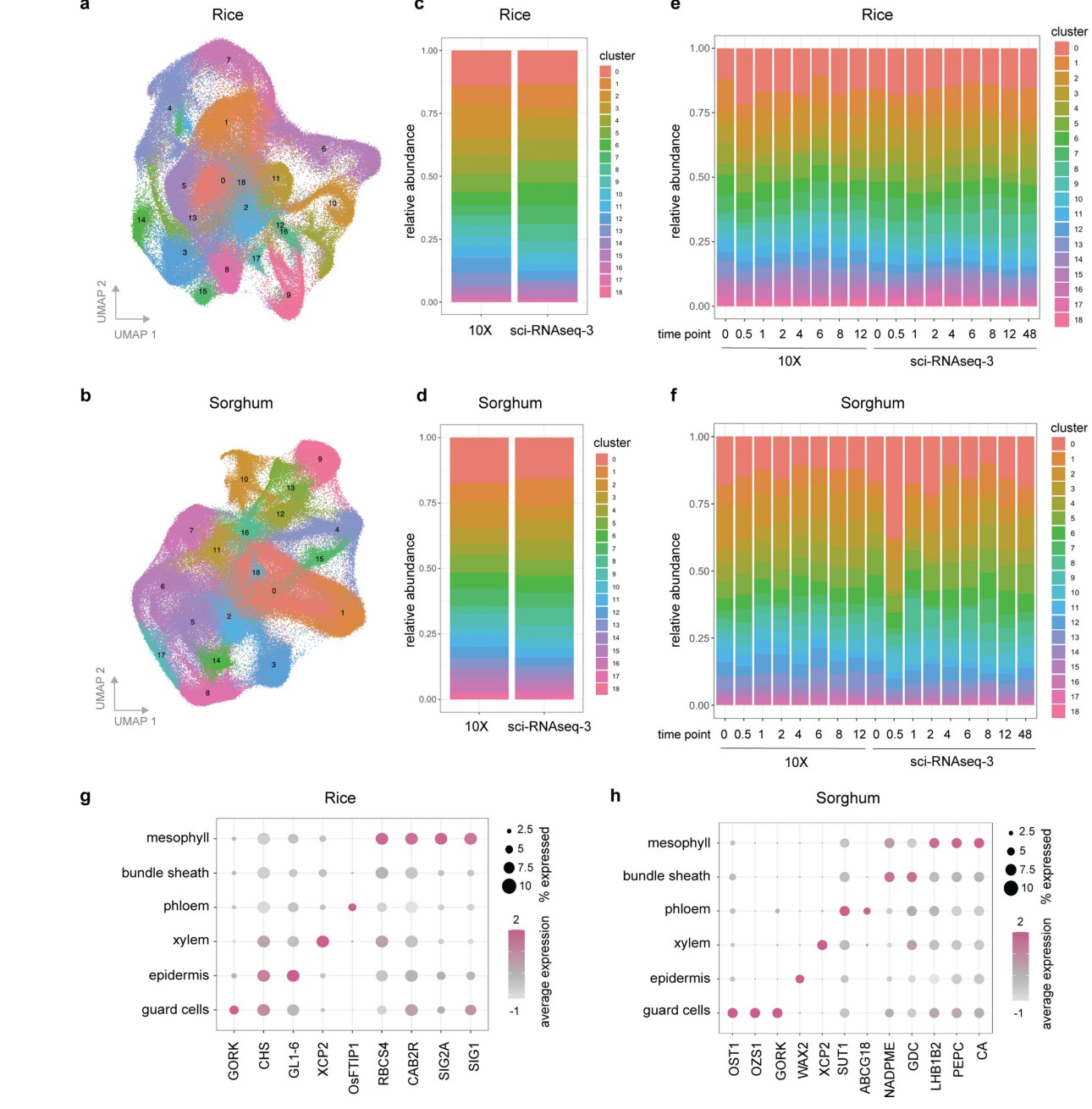

**Extended Data Fig. 2 | Single-nucleus atlases for gene expression in rice and sorghum shoots during de-etiolation. a,b**, UMAP of transcript profiles from single nuclei across rice (**a**) and sorghum (**b**), across all time points tested. **c,d**, Each cluster in rice (**c**) and sorghum (**d**) contained nuclei sequenced either by 10X or by sci-RNA-seq3 methods. **e,f**, Similarly, each cluster in rice (**e**) and sorghum (**f**) contained nuclei sequenced from each time point assayed. **g,h**, Transcript abundance from marker genes in cell types of rice (**g**) and sorghum (**h**).

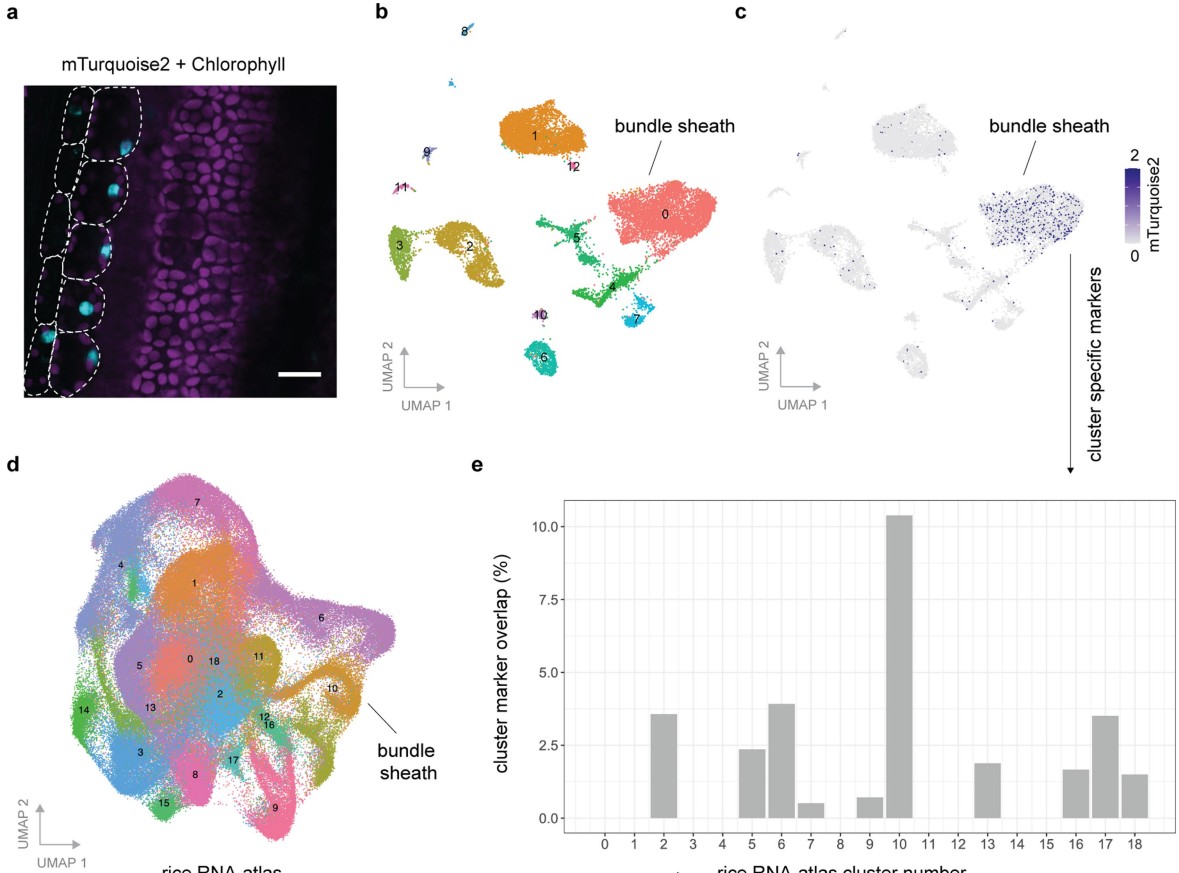

**Extended Data Fig. 3 | Bundle-sheath transgenic marker line identifies the bundle-sheath cluster in the rice transcriptional atlas. a**, Confocal laser scanning microscopy image of a rice bundle-sheath marker line expressing nuclear-localized mTurquoise2 driven by the bundle-sheath-specific Zj*PCK* promoter. Fluorescent signal from chlorophyll is indicated in magenta. The outline of bundle-sheath cells is indicated with a dotted line (scale = 20 μm). Similar expression patterns were observed for mTurquoise2 across 3 independent transgenic lines. **b**, Clustered single nuclei transcript profiles from the rice line expressing mTurquoise2 driven by the bundle-sheath-specific Zj*PCK* promoter. **c**, mTurquoise2 expression in the transgenic line. **d**, Cluster of the rice transcriptional atlas (de-etiolation data). **e**, Per cent overlap of top cluster markers shared between the bundle-sheath transgenic line (**c**) and the rice transcriptional atlas (**d**).

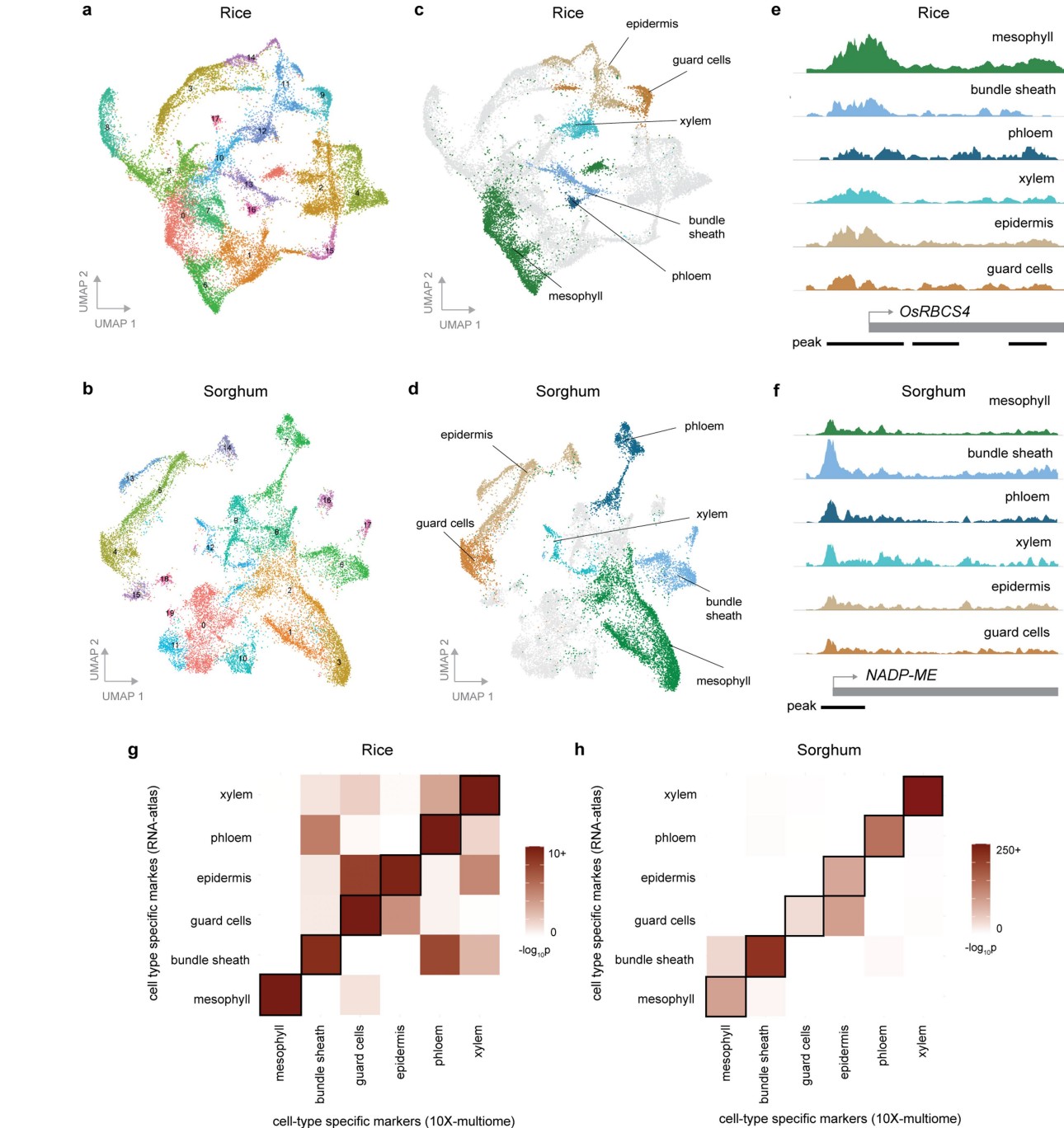

**Extended Data Fig. 4 | Identifying cell types in rice and sorghum 10X Multiome (RNA + ATAC) datasets. a,b**, UMAP clustering of 22,154 rice nuclei (**a**) and 20,169 sorghum nuclei (**b**) sequenced using the 10X Multiome workflow. **c,d**, Cell transcriptional identities assigned to each cluster in rice (**c**) and sorghum (**d**). **e**, Accessibility for the promoter of *OsRBCS4* in each rice cell type.

**f**, Accessibility of promoter region for *NADP-ME* in each sorghum cell type. **g,h**, Overlap of significant marker genes from cell-types identified within the 10X Multiome dataset with those identified within the transcriptome atlas' dataset for rice (**g**) and sorghum (**h**).

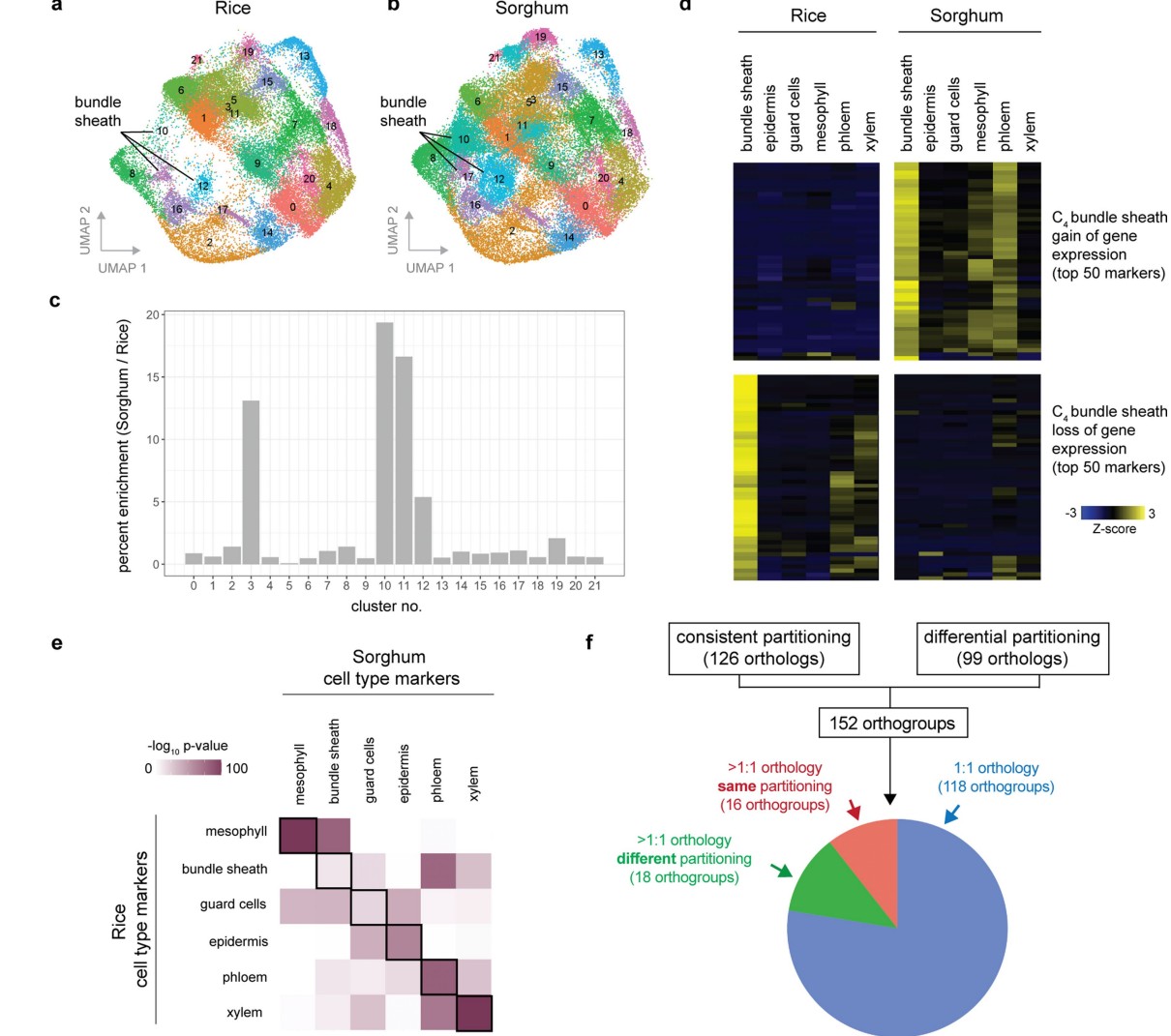

**Extended Data Fig. 5 | Comparison of transcriptional cell identities in rice and sorghum shoots. a**, **b**, UMAP clustering visualizing the single nuclei pan-transcriptomes of rice (**a**) and sorghum (**b**) nuclei 48 h after light exposure. **c**, Fold enrichment of cluster membership found in sorghum relative to that of rice. **d**, Heat map of transcript abundance for bundle-sheath marker genes in rice and sorghum in each cell type 48 h after exposure to light. **e**, Overlap of cell type specific marker genes of rice and sorghum, significance of overlap indicated (log-normalized Fisher's exact test adjusted *P* indicated). **f**, Differentially expressed orthologous gene pairs within mesophyll and bundle-sheath cells of rice and sorghum. Genes fall into two categories, those consistently partitioned (more highly expressed in the same cell type in both rice and sorghum), and those that are differentially partitioned (swap expression from one cell type to the other). Overall, 152 orthogroups were identified, for which the frequency of higher-order orthology relationships (>1:1 orthology) are indicated.

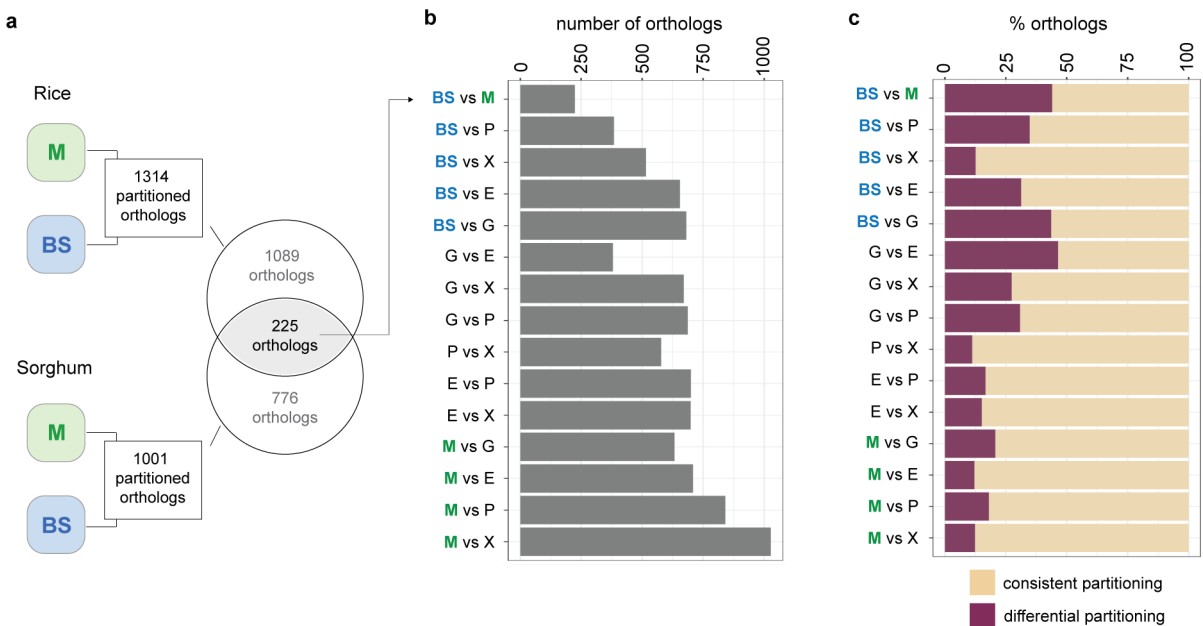

**Extended Data Fig. 6 | Quantifying instances of consistent and differential partitioning across all cell-type pairs. a**, Overlap of orthologous genes found partitioned between mesophyll and bundle-sheath cell types in rice and sorghum. **b**, Quantifying the overlap of orthologous genes partitioned between each possible cell type pair in rice and sorghum. **c**, Percentage of partitioned genes in **b** that are either differentially or consistently partitioned.

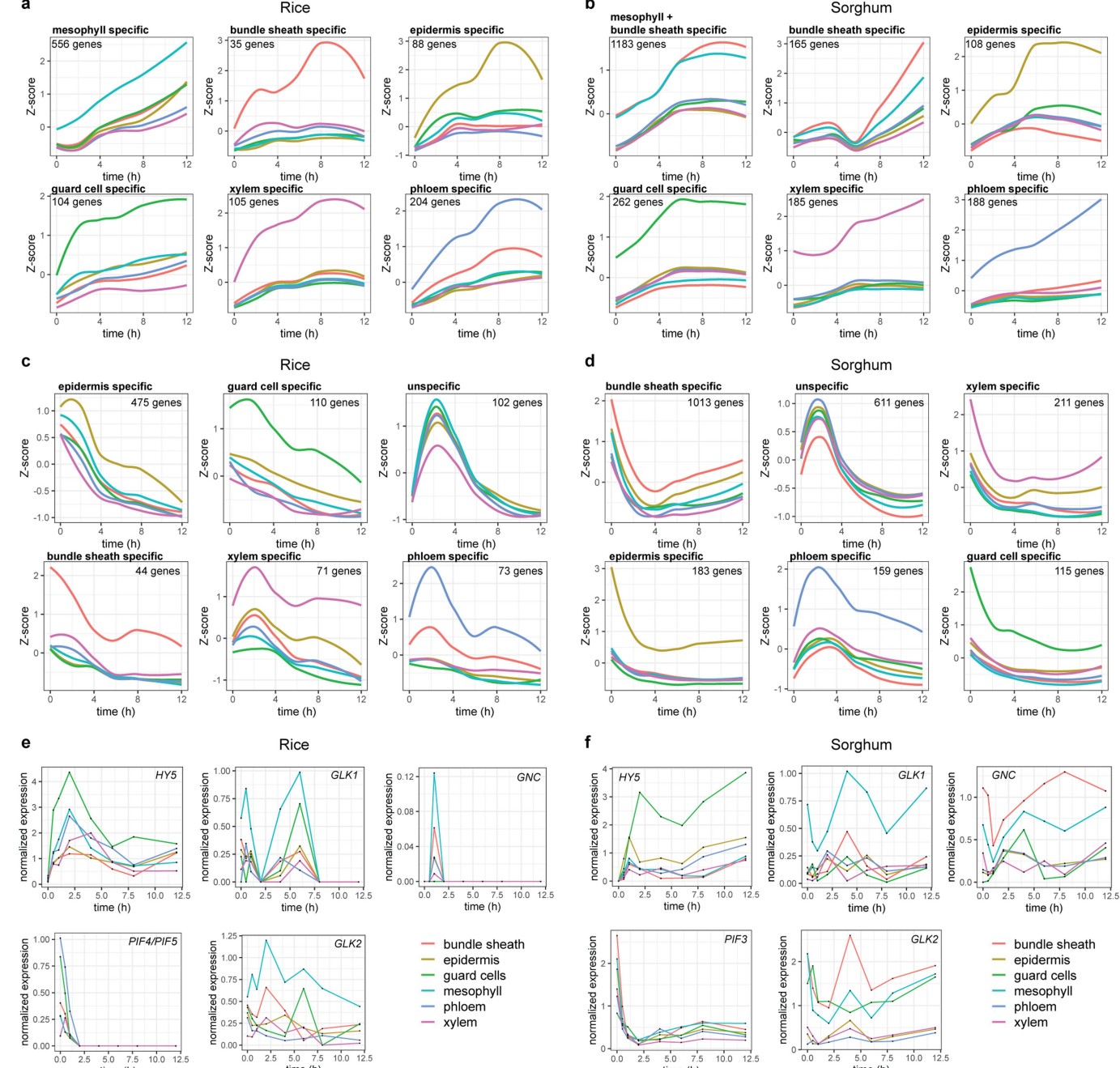

**Extended Data Fig. 7 | Light induces changes in cell-type-specific transcript abundance. a,b**, Activation of transcript abundance of light-responsive genes during the first 12 h of light exposure. Each panel shows *z*-score normalized gene-expression trends unique to each cell type for rice (**a**) and sorghum (**b**). **c,d**, Repression of transcript abundance from light-responsive genes in the first

12 h of exposure to light. Each cluster shows patterns of gene-expression induction unique to each cell type for rice (**c**) and sorghum (**d**). **e,f**, *HY5*, *PIF*, *GLK1 (GOLDEN2-LIKE1)*, *GLK2* and *GNC (GATA NITRATE-INDUCOBLE CARBON-METABOLISM-INVOLVED)* expression in rice (**e**) and sorghum (**f**) during the first 12 h of light exposure across six cell types.

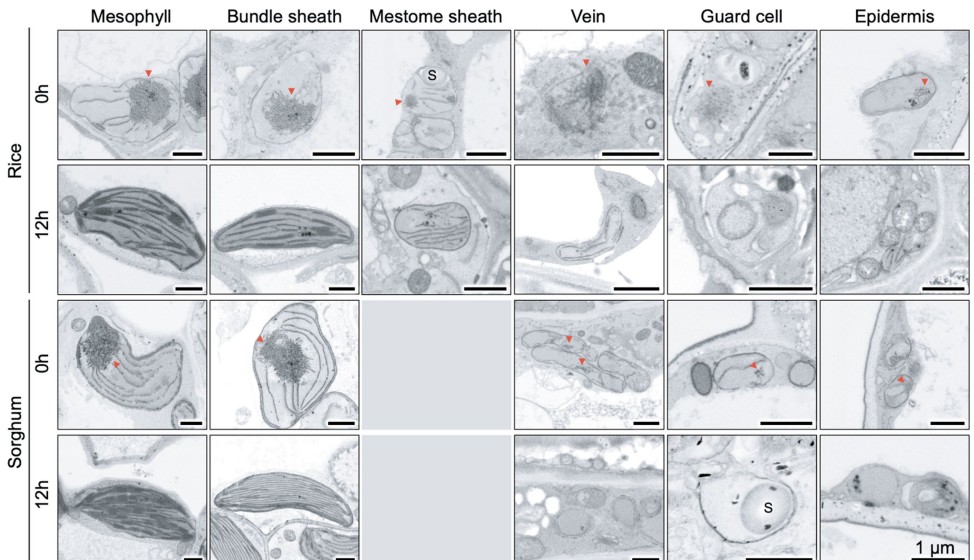

**Extended Data Fig. 8 | Scanning electron micrographs of etioplasts and chloroplasts at 0 h and 12 h after exposure to light in different cell types of rice and sorghum shoots.** Etioplasts are indicated with a red arrowhead. 'S' indicates starch granules. Unlike rice, sorghum does not have a mestome sheath. SEMs were consistent across 3 biological replicates.

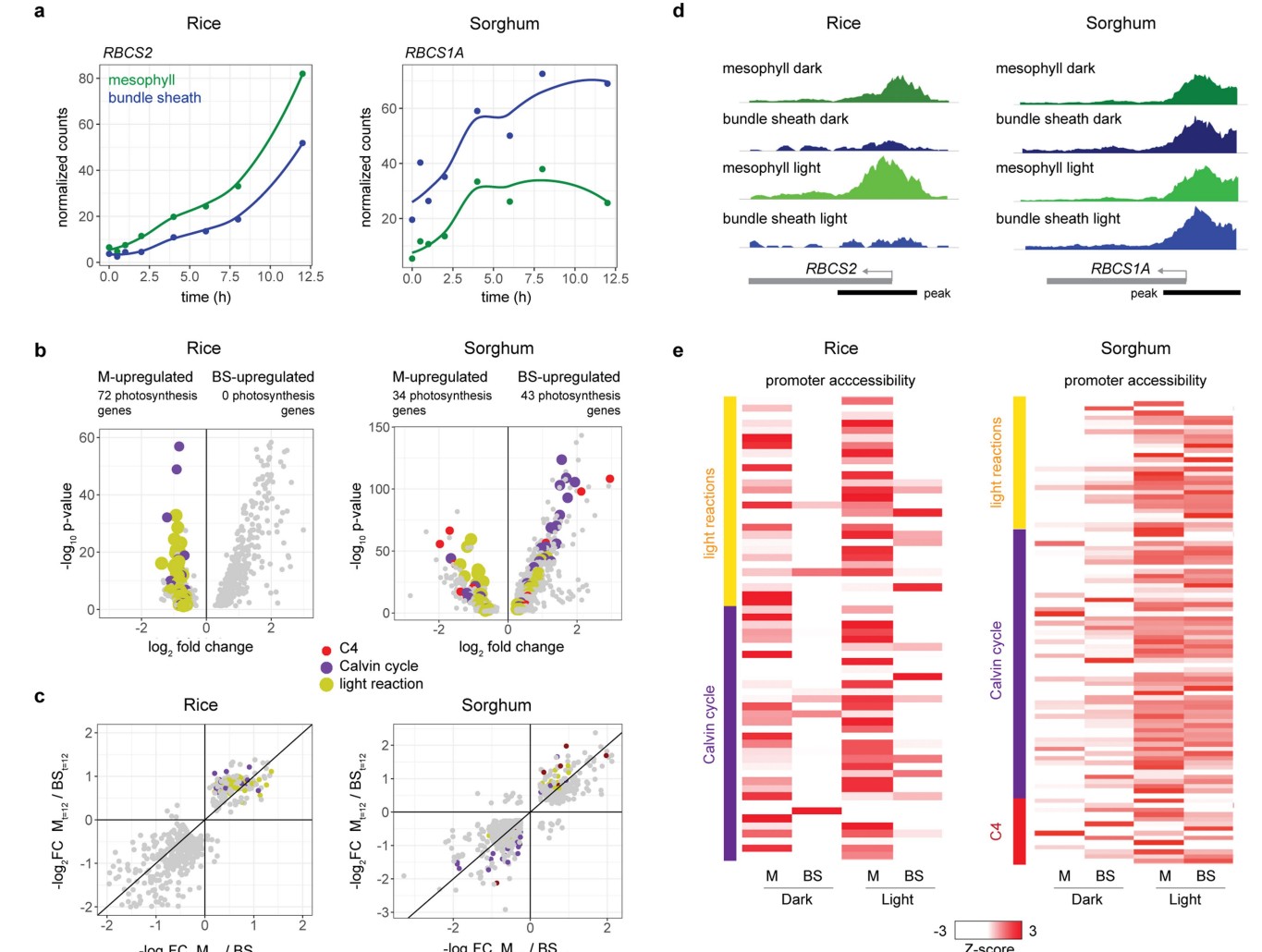

**Extended Data Fig. 9 | Both light exposure and transcriptional cell identity partition photosynthesis gene expression. a**, Transcript abundance of *RBCS2* in rice and *RBCS1A* in sorghum during de-etiolation in mesophyll and bundle-sheath cells of rice and sorghum. **b**, Volcano plots of genes significantly partitioned to either mesophyll or bundle-sheath cell types under light conditions (12-h time point, adjusted *p* < 0.05, likelihood-ratio test). **c**, Differences in fold change gene expression in rice or sorghum genes in the etiolated state (t = 0) vs their expression after 12 h of light exposure between mesophyll and bundle sheath. **d**, Chromatin accessibility in the mesophyll and bundle-sheath promoters of *RBCS2* in rice and *RBCS1A* in sorghum under etiolated and light conditions. **e**, Chromatin-accessibility differences adjacent (+−2,000 bp) to photosynthesis genes at 0 h (dark) and 12 h (light) after light exposure. Genes encoding proteins involved in $C_4$ photosynthesis, the Calvin–Benson–Bassham cycle and the light reactions are shown in red, purple, and yellow, respectively.

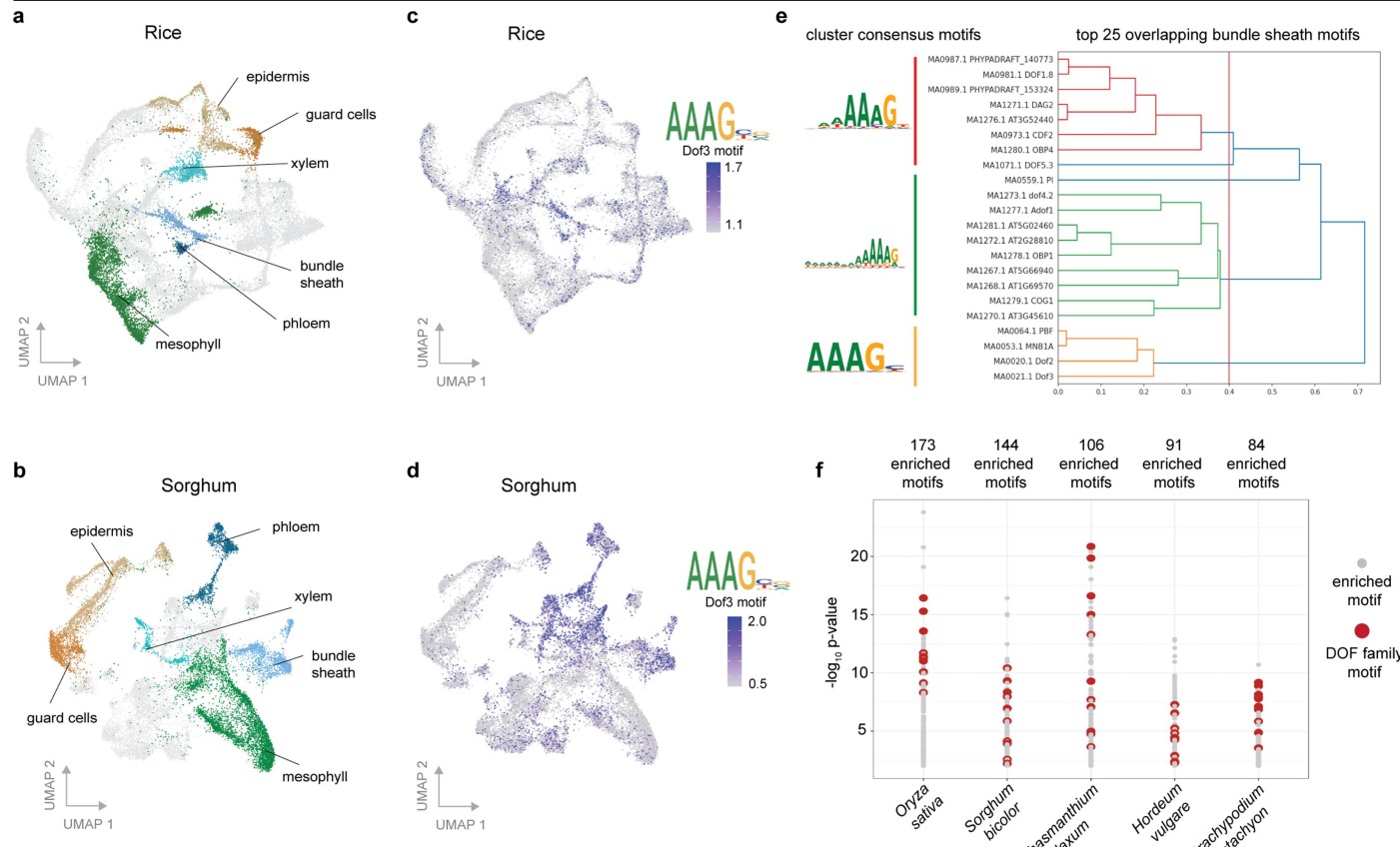

**Extended Data Fig. 10 | Discovering conserved cell-type-specific *cis*-elements across species. a**,**b**, 10X Multiome UMAPs of rice (**a**) and sorghum (**b**). **c**,**d**, Dof3 motif prevalence within accessible chromatin is restricted to bundle-sheath and phloem clusters in rice (**c**) and sorghum (**d**). **e**, For each species, the top 25 most significantly enriched motifs within the bundle sheath were overlapped, and clustered using TOBIAS. A threshold of 0.4 was used to find consensus

motifs (indicated on left). **f**, Number and log-normalized adjusted p-values of enriched *cis*-elements in rice (*Oryza sativa*) and sorghum (*Sorghum bicolor*) within genes that are consistently partitioned to the bundle-sheath cell type in both species (Rank sum test). In addition, enriched *cis*-elements in homologues of bundle-sheath-specific rice genes in the C₃ grasses *Chasmanthium laxum*, *Hordeum vulgare*, and *Brachypodium distachyon* are indicated.

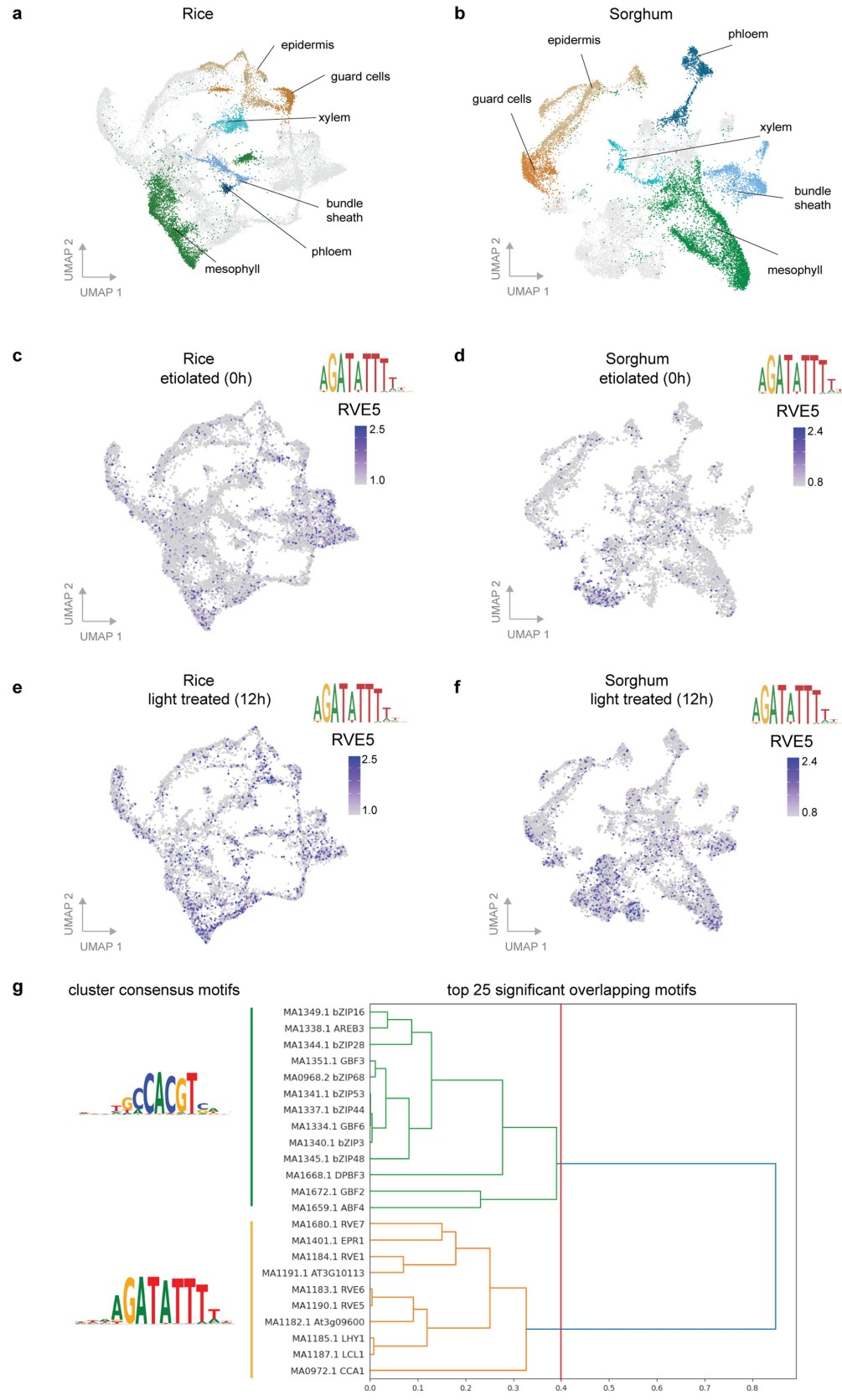

**Extended Data Fig. 11 | Discovering conserved light-responsive *cis*-elements across species. a**,**b**, 10X Multiome UMAPs of rice (**a**) and sorghum (**b**). **c**–**f**, RVE5 motif prevalence within accessible chromatin under etiolated conditions in rice (**c**) and sorghum (**d**), and light conditions in rice (**e**) and sorghum (**f**).

**g**, For each species, the top 25 most significantly enriched motifs were overlapped, and clustered using TOBIAS. A threshold of 0.4 was used to find consensus motifs.

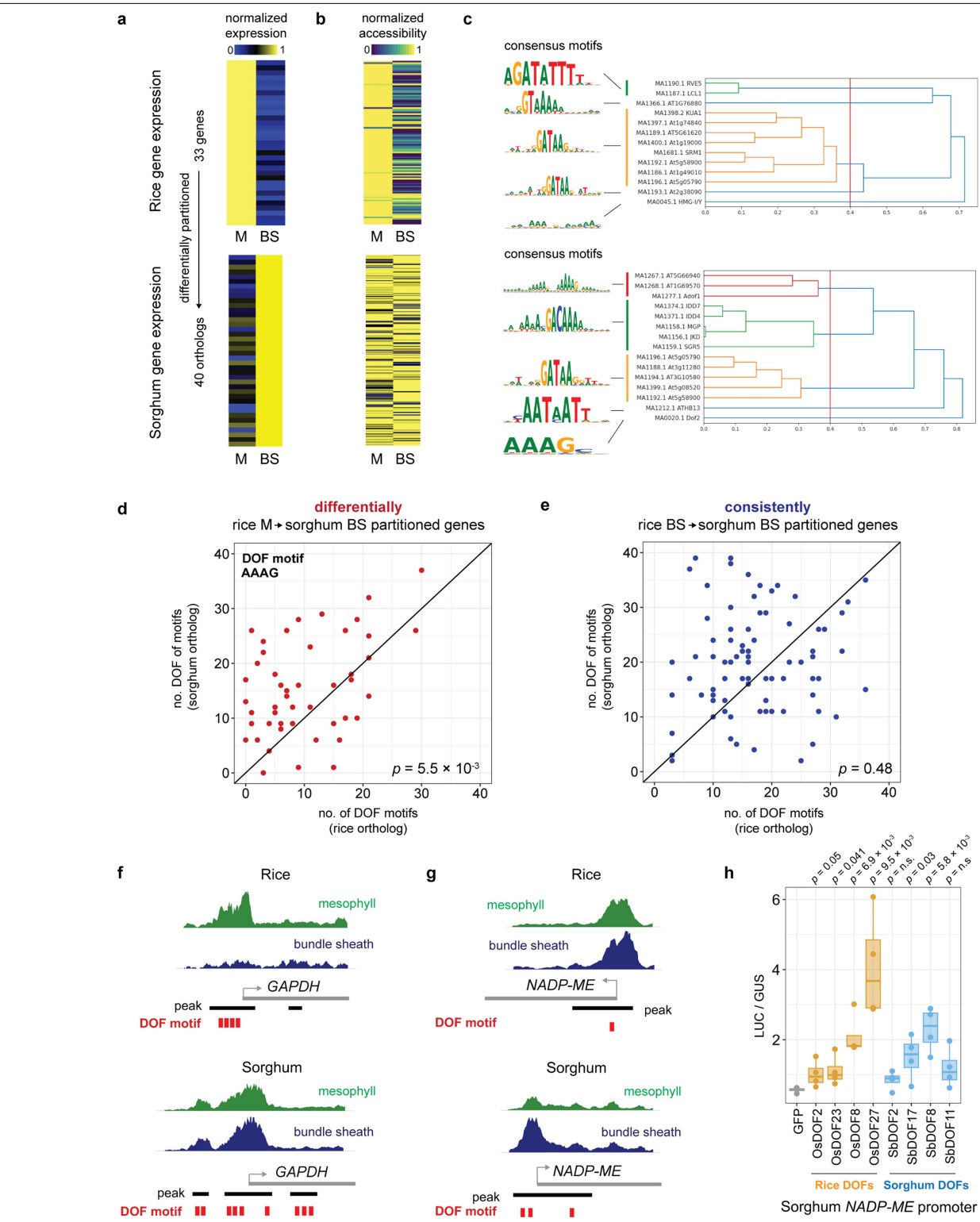

**Extended Data Fig. 12 | Discovering *cis*-elements underlying differentially partitioned genes. a**, Gene-expression heat maps of differentially partitioned genes in rice and sorghum 10X Multiome data. **b**, Accessible chromatin associated with these expression patterns. **c**, Enriched motifs among accessible chromatin, clustered using TOBIAS to find consensus motifs (indicated on left). **d**, Number of DOF motifs within accessible chromatin of differentially partitioned orthologues, two-sided binomial *p* indicated. **e**, Number of DOF motifs within accessible chromatin of consistently bundle-sheath partitioned orthologues, two-sided binomial *p* indicated. **f**, Chromatin accessibility in mesophyll (green) and bundle sheath (blue) for *GAPDH*. Sequences within accessible chromatin were analysed for DOF motifs. **g**, Chromatin accessibility in mesophyll (green) and bundle sheath (blue) for *NADP-ME*. DOF motifs shown in red. **h**, Transactivation of sorghum *NADP-ME* promoter by DOF transcription factors from rice (orange) and sorghum (blue; one-sided Welch's t-test *p* indicated, n = 4 biological replicates, boxes indicate 25th, median and 75th quartiles, whiskers extend to the outermost value within 1.5× of interquartile range, assay repeated 3 times independently with similar results).

# Reporting Summary

## Statistics

For all statistical analyses, confirm that the following items are present in the figure legend, table legend, main text, or Methods section.

| n/a | Confirmed | |
|---|---|---|
| ☐ | ☒ | The exact sample size (*n*) for each experimental group/condition, given as a discrete number and unit of measurement |
| ☐ | ☒ | A statement on whether measurements were taken from distinct samples or whether the same sample was measured repeatedly |
| ☐ | ☒ | The statistical test(s) used AND whether they are one- or two-sided *Only common tests should be described solely by name; describe more complex techniques in the Methods section.* |
| ☐ | ☒ | A description of all covariates tested |
| ☒ | ☐ | A description of any assumptions or corrections, such as tests of normality and adjustment for multiple comparisons |
| ☒ | ☐ | A full description of the statistical parameters including central tendency (e.g. means) or other basic estimates (e.g. regression coefficient) AND variation (e.g. standard deviation) or associated estimates of uncertainty (e.g. confidence intervals) |
| ☒ | ☐ | For null hypothesis testing, the test statistic (e.g. *F*, *t*, *r*) with confidence intervals, effect sizes, degrees of freedom and *P* value noted *Give P values as exact values whenever suitable.* |
| ☒ | ☐ | For Bayesian analysis, information on the choice of priors and Markov chain Monte Carlo settings |
| ☒ | ☐ | For hierarchical and complex designs, identification of the appropriate level for tests and full reporting of outcomes |
| ☐ | ☒ | Estimates of effect sizes (e.g. Cohen's *d*, Pearson's *r*), indicating how they were calculated |

*Our web collection on statistics for biologists contains articles on many of the points above.*

## Software and code

Policy information about availability of computer code

| Data collection | BD Influx Sortware v1.2.0.142 (flow cytometry). Illumina MiSeq control software v3.1.0.13 and NovaSeq 6000 control software v1.6.0/RTA v3.4.4 (sequencing). |
|---|---|
| Data analysis | System: R version 4.2.1, Python 3.10.5<br>Mapping and Assembly: BSgenome 1.64.0, cellranger-3.0.2, cellranger-arc-2.0.1,<br>Anaylsis: ggplot2 3.4.0, Seurat 4.3.0, Signac 1.9.0, EDASeq 2.30.0, DESeq2 1.36.0<br>Other code and required packages can be found at https://github.com/joey1463/C3-C4.git |

For manuscripts utilizing custom algorithms or software that are central to the research but not yet described in published literature, software must be made available to editors and reviewers. We strongly encourage code deposition in a community repository (e.g. GitHub). See the Nature Portfolio guidelines for submitting code & software for further information.

## Data

Policy information about availability of data

All manuscripts must include a data availability statement. This statement should provide the following information, where applicable:
- Accession codes, unique identifiers, or web links for publicly available datasets
- A description of any restrictions on data availability
- For clinical datasets or third party data, please ensure that the statement adheres to our policy

> Raw and processed data, including assembled atlases, have been deposited at GEO and are publicly available (GSE248919).  Chlorophyll measurement raw data have been deposited at Mendeley. (doi: 10.17632/6xmsdg9xcr.1). R scripts used for analyses presented in this manuscript are published on github at https://github.com/joey1463/C3-C4.git.

## Research involving human participants, their data, or biological material

Policy information about studies with human participants or human data. See also policy information about sex, gender (identity/presentation), and sexual orientation and race, ethnicity and racism.

| | |
|---|---|
| Reporting on sex and gender | not applicable |
| Reporting on race, ethnicity, or other socially relevant groupings | not applicable |
| Population characteristics | not applicable |
| Recruitment | not applicable |
| Ethics oversight | not applicable |

Note that full information on the approval of the study protocol must also be provided in the manuscript.

# Field-specific reporting

Please select the one below that is the best fit for your research. If you are not sure, read the appropriate sections before making your selection.

☒ Life sciences          ☐ Behavioural & social sciences          ☐ Ecological, evolutionary & environmental sciences

For a reference copy of the document with all sections, see nature.com/documents/nr-reporting-summary-flat.pdf

# Life sciences study design

All studies must disclose on these points even when the disclosure is negative.

| | |
|---|---|
| Sample size | No sample size calculation was performed. The number of nuclei we sequenced per time point allowed us enough coverage to annotate cell types and perform required statistical analyses to detect differentially expressed genes and accessible chromatin. |
| Data exclusions | No significant data was excluded from our study. Only genes with low read counts, or nuclei with low UMIs, were removed during normal processing of single-nuclei sequencing data. |
| Replication | Single nuclei sequencing data presented in this study was replicated in the following ways:<br>- For our assay of transcriptional responses to photomorphogenesis, multiple time points were assayed (9 time points).<br>- We sequenced tissue resulting from this time course using both 10X and sci-RNA-seq3 using different plant populations (technical replication, biological replication).<br>- For our multiomic assay, we replicated these with 2 distinct biological replicates and 2-4 technical replicates (technical replication, biological replication). All replication events were successful.<br>Validation experiments in this study were replicated in the following ways:<br>- For our transactivation assay testing DOF transcription factor activity, we performed the experiment 3 times independently with similar results, each with 4 biological replicates. We report only one experiment.<br>- For our GUS reporter of the minimal SIR reporter, we performed 52 independent transformation events.  Resulting data for all independent lines are presented.<br>- For chlorophyll measurements, we performed 3 biological replicates, and performed the experiment 3 times independently with similar results. |
| Randomization | Plant seedlings were randomized in growth chamber. Additionally, random seedling samples (n > 10) were taken at each time point observed. Random subpopulations of nuclei were selected either through FACS sorting or random passage through the sci-RNA-seq3 workflow. |
| Blinding | Blinding was not relevant in our study. Specifically, photomorphogenesis will turn white tissue visibly green. Thus, throughout our experimental protocol it is possible to identify which sample was light treated. |

# Reporting for specific materials, systems and methods

We require information from authors about some types of materials, experimental systems and methods used in many studies. Here, indicate whether each material, system or method listed is relevant to your study. If you are not sure if a list item applies to your research, read the appropriate section before selecting a response.

## Materials & experimental systems

| n/a | Involved in the study |
|-----|----------------------|
| ☒ | ☐ Antibodies |
| ☒ | ☐ Eukaryotic cell lines |
| ☒ | ☐ Palaeontology and archaeology |
| ☒ | ☐ Animals and other organisms |
| ☒ | ☐ Clinical data |
| ☒ | ☐ Dual use research of concern |
| ☐ | ☒ Plants |

## Methods

| n/a | Involved in the study |
|-----|----------------------|
| ☒ | ☐ ChIP-seq |
| ☒ | ☐ Flow cytometry |
| ☒ | ☐ MRI-based neuroimaging |

## Dual use research of concern

Policy information about dual use research of concern

### Hazards

Could the accidental, deliberate or reckless misuse of agents or technologies generated in the work, or the application of information presented in the manuscript, pose a threat to:

| No | Yes | |
|-----|-----|---|
| ☒ | ☐ | Public health |
| ☒ | ☐ | National security |
| ☒ | ☐ | Crops and/or livestock |
| ☒ | ☐ | Ecosystems |
| ☒ | ☐ | Any other significant area |

### Experiments of concern

Does the work involve any of these experiments of concern:

| No | Yes | |
|-----|-----|---|
| ☒ | ☐ | Demonstrate how to render a vaccine ineffective |
| ☒ | ☐ | Confer resistance to therapeutically useful antibiotics or antiviral agents |
| ☒ | ☐ | Enhance the virulence of a pathogen or render a nonpathogen virulent |
| ☒ | ☐ | Increase transmissibility of a pathogen |
| ☒ | ☐ | Alter the host range of a pathogen |
| ☒ | ☐ | Enable evasion of diagnostic/detection modalities |
| ☒ | ☐ | Enable the weaponization of a biological agent or toxin |
| ☒ | ☐ | Any other potentially harmful combination of experiments and agents |

## Plants

**Seed stocks**

We used wild type Oryza sativa spp Japonica cultivar Kitaake, and Sorghum Bicolor BTx623

**Novel plant genotypes**

Oryza sativa spp Japonica cultivar Kitaake was transformed with a construct tagging bundle sheath nuclei with mTurquoise2. Transformation method was agrobacterium mediated rice transformation. We analyzed 5 independent transgenic single copy lines. Experiments were performed homozygous T2 and T3 seeds. Additionally,
Oryza sativa spp Japonica cultivar Kitaake was transformed with a construct tagging bundle sheath nuclei with the minimal sulfite reductase (SIR) promoter driving GUS expression. Here, 23 and 29 independent T0 plants for two different construct types were analyzed.

**Authentication**

The presence of the transgene was confirmed using southern blots, PCR and confocal microscopy.

