## [Peer Review File · Nature]

Manuscript Title: Exaptation of ancestral cell identity networks enables C4 photosynthesis

Reviewer Comments & Author Rebuttals

Reviewer Reports on the Initial Version:

Referees' comments:

Referee #1 (Remarks to the Author):

This is an exciting and well written paper that uses numerous genome-wide transcriptional and chromatin accessibility assays to investigate how specific tissues in the leaves acquire new abilities to perform photosynthesis reactions. There are several general evolutionary models for how tissues gain new functions, some relying on innovations of new transacting factors. Here, the authors provide a compelling story based on their data that suggests it's the acquisition of cell-type specific (DOF) transcription factor binding sites in C4 gene promoters that ties expression of these genes into the pre-existing patterning of transcription factors to enable bundle sheath cell expression. While there is still room for alternative interpretations, this work provides a conceptually intriguing idea along with some technically impressive data.

I have only a few comments that can be addressed with changes to the text and figures and perhaps some different data analysis; none require new data.

Lines 200-210:

The proposed similarity of cell types is dependent on the number and specificity of the markers used, and I could see that if a cell type was poorly defined at a gene-expression level, that this plot could be unintentionally misleading. For example, xylem looks to be highly conserved but guard cells not. Is this really that guard cell gene expression varies widely across evolution (despite morphological and physiological similarities) or is it that markers of guard cells used in the analysis (e.g. transporters) may not be all that cell-type specific? In the context of this paper, I understand that the key point is the BS cell comparison, but providing some of the biological context here is important. Please provide some numbers to give the reader a sense of how many genes were considered in making this chart. This would be best in the text, but if space restricts this, it could go into the methods.

Line 305-320:

This section overstates the degree to which we know a cis-regulatory "code" defines gene expression. Certainly, cis-sequences contribute (and may even be the majority of the contribution), but this section and Figure 4d present only the top motifs. While this might be fine in yeast or another system where there are relatively small TF families and for which there are extensive experimental binding data about specific TFs and verifiable targets, this is not the situation with any plant. Revising the text to recognize this nuance would not remove any impact of this work and would better represent what is experimentally supported.

Figures:

In general, the visual elements are laid out in an aesthetically appealing way. Part of the appeal is that they are not overcrowded, but this does make it essential that sections are well explained in the legends.

It may be a demand of journal style, but I found it very challenging to interpret the visuals from the associated legends. If possible, additional text in the legends would make this much more readable.

Some specifics:

Fig 1

1a: Define what OH corresponds to

1b: add annotations to micrographs to point out the differential thylakoid stacking described in the text.

1c-d, is color supposed to indicate cell type and are the same colors used for each species?

1e, Same section +/- chlorophyll? What is inclusion of this channel supposed to tell me? Arrows or something to annotate image would help.

Fig 2

UMAP is defined here, but not in Fig 1 (move to Fig 1).

2a, implication is that black circles in sorghum are underrepresented in rice, but this isn't explicitly stated. It also wasn't entirely obvious that "Pan-transcriptome" means that the nuclei from both species were mapped together and then presented separately, so if that is what was done, stating it directly would again be helpful.

Fig 3

3h, please write the source of data for these experiments—multiome experiments?

The abstract is clear and well written.

Referee #2 (Remarks to the Author):

Summary of key results: In this manuscript, the authors use single-cell RNAseq combined with ATAC-seq to determine the mechanism behind the transcriptional re-wiring that was key in the evolution of C4 photosynthesis. They find that the C4 plant sorghum co-opted an existing regulatory network in order to activate photosynthesis genes within the bundle sheath cells, which was an essential innovation for this more efficient form of photosynthesis.

I found this manuscript very interesting, with elegant experimental design and an exciting and scientifically robust result that offers insight into one of the most important biological pathways in plants. The methods used were well-suited to study the question presented by the authors, the analyses were appropriate, and the results were all presented very clearly in the figures.

I do think that the authors could do more to make their manuscript more accessible to non-plant scientists, who may have trouble recognizing the significance of the final result if they don't fully understand how the localization of photosynthesis to different cell types is important. The authors do discuss this idea in the introduction, but my suggestion would be to also add a figure to the beginning of the manuscript giving a broad overview of C3 versus C4 photosynthesis, to help orient readers who may be much less familiar with this process. This would provide a nice set-up to Figure 5D, where the main findings are summarized.

I also have a few other minor comments/questions:

In lines 101-102, it is mentioned that sorghum and rice may share molecular signatures with the many other species derived from their last common ancestor. There's a nice paper from back in 2010 that did ancestral karyotype reconstruction which shows how much of the ancestral genome (and genome structure) both sorghum and rice still share with each other and the grass ancestor, which I think lends even more weight to this argument that these species are an excellent model for the work being done here. (Murat et al. 2010. Ancestral grass karyotype reconstruction unravels new mechanisms of genome shuffling as a source of plant evolution. *Genome Research* 20: 1545-1557.)

In lines 211-214: The phrase "gained genes" makes it sound like sorghum gained new gene copies or novel gene sequences, but I don't think that is what you mean since (as I understand it) you are only looked at expression in genes with orthologs in rice. This should be clarified.

In line 573 of the Methods, it is stated that "if multiple orthologs for a gene were found across species, only one was retained." Does this mean that you also removed any expression data if transcripts mapped to a duplicate gene copy, and only retained counts for one copy? Is this definitely controlling for the possibility that some increases in expression could come from copy number variants (and not just gene regulation)?

Referee #3 (Remarks to the Author):

This is an important and exciting study that is well designed, well executed, and well described. Though some technical elements were outside my area of expertise, I nonetheless was able to follow along quite well due to their clear explanations. Their results are important for our understanding of C4 photosynthesis evolution, but also evolutionary innovation more broadly.

The basic question this paper asks is how, during C4 evolution, the bundle sheath cells become recruited as the major carbon-fixing cells of the plant leaf. Through analyses of single-cell RNA expression of sorghum (C4) and rice (C3), they discover that the core set of Calvin-cycle genes as well as a key decarboxylation gene have acquired key cis-element motifs in Sorghum that are part of a bundle-sheath identity network, regulated by a DOF family transcription factor. Their inference is that this cell-identity network is ancestral, because it also regulates bundle sheath development in

rice. Thus, sorghum utilized a pre-existing bundle sheath identity module to incorporate cell-specific expression of key photosynthesis genes that is essential for a functional C4 cycle. They contrast this result to other papers focused in eudicots, which suggest that the expression of the transcription factors themselves change spatially, and cis-elements are conserved.

One important contribution from this paper is the identification of bundle-sheath (BS) specific marker genes in C3 plants- this apparently required transformation and fluorescent labeling of BS nuclei with a promoter of phosphoenolpyruvate carboxykinase (PCK). This is not my area of expertise, but as I thought more about it, it became unclear to me why they took this additional step, as they are simply sorting nuclei according to PCK expression. So they are already using PCK as a “marker gene” of C3 BS, based on Nomura et al. 2005. Wouldn't they get the same result if they had used PCK as a marker gene in the same way they used other known localized genes in mesophyll, cuticle, xylem, etc, i.e., without doing the transformation experiment? The rationale could be better explained here.

My only concern with the paper might be an over-confidence in what is inferred to be “ancestral”, as the experiment with sorghum and rice is really a two-taxon statement – there is literally no way to infer any directionality to differences found between two species. The authors infer that the sorghum state is derived because we know that C4 is derived – it is not terrible logic, but they should carefully lay out their reasoning for the readers. One resource that is readily available to them is a good sampling of annotated Poaceae genomes accessible via Phytozome- including important C3 PACMAD species like *Chasmanthium*. While not nearly as informative as the ATAC-seq analyses performed in their experiments, the authors could very quickly target some of their BS-specific genes and examine upstream regions for DOF-binding motif enrichment across a broader collection of grasses. I'm not insistent but I think a quick analysis along these lines would lend support to their inferences of the DOF-BS identity network being ancestral.

Smaller point about their title- is this really a ‘re-wiring’? It seems the opposite to me- conservation of an old wiring, but adding in new components.

Referee #4 (Remarks to the Author):

This manuscript describes the generation of complementary high-quality datasets in two cereal species, rice and sorghum, to study the molecular signaling cascades controlling C3 and C4 photosynthesis. Based on high-resolution transcript and accessible chromatin profiling using state-of-the-art single-cell -omics methods, gene expression and chromatin state dynamics are characterized for seedlings undergoing photomorphogenesis. Comparative analysis using known and novel marker genes is used to annotate different clusters, representing distinct cell types (incl. validation of a bundle sheath reporter line). Comparative analysis using gene orthology information is subsequently used to characterize similarities and differences in gene expression, focusing on mesophyll and bundle sheath, two cell types playing a key role in both photosynthesis pathways. The analysis of expression divergence between both species allows the delineation of a set of genes showing differential partitioning, very likely contributing to the differences between the C3 and C4

pathways. Chromatin accessibility analysis during dark and light confirms major differences in the regulation of photosynthesis genes, suggesting that cell identity and associated cis-elements are responsible for the partitioning of photosynthesis genes. Taken together, the authors propose a model where a major difference between rice (C3) and sorghum (C4) is the acquisition of DOF cis-elements in sorghum photosynthesis genes expressed in bundle sheath.

The manuscript starts with an excellent introduction outlining the relevance and major differences between C3 and C4 photosynthesis in plants. The transcript and accessible chromatin profiling results are well described. The Figures and Extended Data Figures do a good job to report the major trends in this complex cross-species dataset, though on some occasions the figure legends are very brief to fully understand all results shown in the different panels. While I largely agree with the interpretation of the reported gene expression differences between rice and sorghum, I am missing detailed results on the cis-element analysis that fully support the claims that are made in the proposed model. Furthermore, and in contrast to previous reports by the same authors, no validation experiments were performed to test some of the hypotheses that are resulting from the proposed model.

Major

1. The authors rely on gene orthology information to perform various comparative analyses between rice and sorghum. Line 572 mentions ortholog conversions were performed in a one-to-one manner, meaning that in some cases, when multiple orthologs are present, only one ortholog was retained. While I acknowledge that for some comparative analyses orthogroups are used, it is unclear how cases where multiple orthologs (e.g. one rice gene has two sorghum orthologs) showing different or opposite expression patterns (e.g. up vs down, or BS vs M for the two sorghum orthologs) are dealt with, and how these affect the presented partitioning results. Please provide more details including counts on the number of one-to-one and more complex orthology relationships.
2. Results reported in Suppl. Tables 10-12 (please add TF family information for these motif identifiers) are insufficient to understand how different TF motifs are in both cell types, for both species. To verify the important claims made in the article about the role of DOF TF motifs in genes swapping expression (lines 338-339), the authors need to report motif frequencies for the different gene sets (Fig 5a) in rice and sorghum. The adjusted p-values reported in the Suppl. Tables are insufficient to verify how many genes, and which genes, have specific TF motifs. The example shown in Fig 5b is not very convincing (the GADPH and ALDOLASE peaks contain DOF motifs in both rice and sorghum, despite the difference in accessible chromatin; the NADP-ME example is better). Furthermore, a limitation of the current study is that no evidence is presented which DOF TFs are controlling BS expression (see comment 4).
3. Following the previous comment, based on the difference in accessible chromatin shown in Extended Data Figure 13a (the BS accessibility in rice is much lower than in sorghum), a difference in the control of accessible chromatin (without a direct role of specific TFs, so potentially epigenetic) could provide an alternative explanation. Validation experiments, for example by perturbation of specific cis-elements to measure the effect on gene expression and/or photosynthetic activity, like the authors did in Singh et al., 2023, are needed to strengthen the proposed model.

4. I suggest the authors include reference <https://pubmed.ncbi.nlm.nih.gov/34783109/> in the Discussion, as this paper performed related experiments in maize, also hinting towards the role of DOF TFs incl. ChIA-PET data linking specific TFs to photosynthesis genes.

5. Several figures have a very brief legend, making it hard to fully understand the different panels.

a. Fig 1C. transcript profiles of which assay are shown here (cfr. Fig1A)?

b. Fig 2f: please extend text legend (too cryptic to understand)

c. Fig 2g: add gene symbols of important or relevant genes (suggestion). Or refer to the corresponding Suppl. Table where these genes are listed.

d. Extended data Figures 4, 6: please add cell type annotation for the different clusters. For example, in Extended Data Figure 4, add that cluster 10 is BS.

e. Extended Data Figure 7: please check the legend, there is an error in part (c).

6. Fig 4e and Extended Data Figure 11-12: unclear which motifs are specific for which cell type in which species. Why are the top 25 motifs compared (do the conclusions change when comparing more/all enriched motifs)? For specific photosynthetic gene sets discussed in the article, please provide gene – TF motif info (in accessible peaks), as apart from the three genes discussed in the text, no info for other photosynthesis genes is given.

Minor

7. Line 329-330: Given the previous sentences reported specific genes, the start of this sentence does not make sense. I assume you want to identify motifs associated with (differential?) accessible chromatin peaks for these genes? Please revise.

8. Line 335: move to Methods. As permutation statistics were applied here, how was the statistical significance of enriched motifs determined for the previously reported motif results. Why were different methods used? The Methods section on Cis-element analyses is hard to understand (also see next comment) and hard to reproduce. Sharing source code or example command lines how these tools were executed, on which datasets, would help understanding what was done.

9. Line 622: Unclear what you mean with “we calculated position frequency matrices’. You converted PWMs to position frequency matrices? Or you mapped known JASPAR (I assume JASPER is a typo) to the rice and sorghum genome? Or you did de novo motif finding to learn position frequency matrices? Please revise.

10. Line 639: “ignoring areas that were inside gene bodies.” Does this mean that cis-regulatory elements in introns or UTR are all discarded? Given the increasing evidence that also these gene regions are important in controlling gene expression, this seems like a harsh filtering step.

Author Rebuttals to Initial Comments:

Editors comments:

Your manuscript, "Rewiring of ancestral cell identity networks enables C4 photosynthesis", has now been seen by 4 referees, whose comments are attached below. While they find your work of potential interest, as do we, they have raised important concerns that in our view need to be addressed before we can consider publication in Nature.

We would specifically like to draw your attention to comments made by reviewer #4 regarding experimental validation of the most salient results, and comments made by all reviewers regarding more caution and nuance in the conclusions drawn overall.

Response: Thank you for your valuable feedback on our manuscript. We acknowledge the points raised by the editor and Referee #4 regarding the necessity of experimental validation. In response, we have incorporated data from stable transgenic lines demonstrating that DOF binding sites regulate expression in bundle sheath cells in rice. Additionally, we have included effector assays that demonstrate that specific members of the DOF transcription factor family are sufficient to activate the expression of target genes containing the DOF binding site. Please refer to the detailed explanations below.

Referees' comments:

Referee #1 - plant development (Remarks to the Author):

This is an exciting and well written paper that uses numerous genome-wide transcriptional and chromatin accessibility assays to investigate how specific tissues in the leaves acquire new abilities to perform photosynthesis reactions. There are several general evolutionary models for how tissues gain new functions, some relying on innovations of new transacting factors. Here, the authors provide a compelling story based on their data that suggests it's the acquisition of cell-type specific (DOF) transcription factor binding sites in C4 gene promoters that ties expression of these genes into the pre-existing patterning of transcription factors to enable bundle sheath cell expression. While there is still room for alternative interpretations, this work provides a conceptually intriguing idea along with some technically impressive data.

I have only a few comments that can be addressed with changes to the text and figures and perhaps some different data analysis; none require new data.

Response: Thank you very much for the careful analysis and positive views on this work. We respond to your specific points below.

Lines 200-210: The proposed similarity of cell types is dependent on the number and specificity of the markers used, and I could see that if a cell type was poorly defined at a gene-expression level, that this plot could be unintentionally misleading. For example, xylem looks to be highly conserved but guard cells not. Is this really that guard cell gene expression varies widely across evolution (despite morphological and physiological similarities) or is it that markers of guard cells used in the analysis (e.g. transporters) may not be all that cell-type specific? In the context of this paper, I understand that the key point is the BS cell comparison, but providing some of the biological context here is important. Please provide some numbers to give the reader a sense of how many genes were considered in making this chart. This would be best in the text, but if space restricts this, it could go into the methods.

Response: Thank you for this very good point. As requested, we now list the cell type specific genes within each species considered in making this chart in Supplementary Table 5 (now referred to in the main text on line 207). We have also referred to the numbers for the bundle sheath in the main text (lines 209-210) to give more context and have given an overview of numbers in the methods sections (lines 671-676). In our analysis, we set a minimum statistical threshold (adjusted p-value < 0.01) when calling cell-type specific markers. Our logic is that any gene that surpasses this threshold is considered cell type specific. In this way, the strength of the cell type specific expression found in guard cells was statistically comparable to the cell type specific expression in any other cell type. We also note that the number of cell type markers found for each cell type is comparable. For example, for guard cells we find 393 and 345 specific markers for rice and sorghum respectively, which is similar to the average number of markers found amongst other cell types (381 and 462 for rice and sorghum respectively).

Line 305-320: This section overstates the degree to which we know a cis-regulatory “code” defines gene expression. Certainly, cis-sequences contribute (and may even be the majority of the contribution), but this section and Figure 4d present only the top motifs. While this might be fine in yeast or another system where there are relatively small TF families and for which there are extensive experimental binding data about specific TFs and verifiable targets, this is not the situation with any plant. Revising the text to recognize this nuance would not remove any impact of this work and would better represent what is experimentally supported.

Response: *We agree. In line with the reviewer's suggestion, we have now presented more nuance in the main text explaining our findings. We now state on lines 315-321: “Cis-regulatory DNA sequences play an important role in driving the patterning of gene expression. Therefore, we next searched for cis-elements that underlie the observed cell identity- and light-dependent patterns of gene expression in rice and sorghum. When regions of open chromatin specific to each cell type were assessed for over-represented transcription factor binding sites, we found dozens of enriched cis-regulatory elements for each cell type (Supplementary Table 10). We compared the 25 most significantly enriched cis-regulatory motifs for each cell type across species.” Additionally, we now refer to all enriched cis-elements listed in Supplementary Table 10 in the figure legend for Figure 4d.*

Figures:

In general, the visual elements are laid out in an aesthetically appealing way. Part of the appeal is that they are not overcrowded, but this does make it essential that sections are well explained in the legends. It may be a demand of journal style, but I found it very challenging to interpret the visuals from the associated legends. If possible, additional text in the legends would make this much more readable.

Response: *Thank you for pointing this out, we have now added more detail and explanations to all figure legends.*

Some specifics:

Fig 1a: Define what OH corresponds to

Response: *This is now explained in the legend.*

1b: add annotations to micrographs to point out the differential thylakoid stacking described in the text.

Response: *We apologise for this omission - this is now done.*

1c-d, is color supposed to indicate cell type and are the same colors used for each species?

Response: *The colours indicate cell type and are the same for each species. We have now added this information in the figure legend to increase clarity.*

1e, Same section +/- chlorophyll? What is inclusion of this channel supposed to tell me? Arrows or something to annotate image would help.

Response: *Apologies for the lack of clarity – we had intended for both channels to help make it clear which cell type was being imaged. We acknowledge that this was not having this affect. We have now included a better description in the legend, and also displayed a section of the image that allows clear identification of bundle sheath cells with both chlorophyll and mTurquoise2 channels overlaid, and have additionally outlined the bundle sheath cells with dotted lines.*

Fig 2: UMAP is defined here, but not in Fig 1 (move to Fig 1).

Response: *We apologise for the omission - this is now done.*

2a, implication is that black circles in sorghum are underrepresented in rice, but this isn't explicitly stated. It also wasn't entirely obvious that “Pan-transcriptome” means that the nuclei from both species were mapped together and then presented separately, so if that is what was done, stating it directly would again be helpful.

Response: *Thank you. We now make clearer in the main text how we generated the pan-transcriptome atlas. Specifically, on lines 186-187 we write: “This was achieved by identifying*

sorghum and rice orthologs and clustering nuclei from both species together.” Additionally, we have also added a more detailed explanation to the figure legend for Figure 2a.

Fig 3: 3h, please write the source of data for these experiments—multiome experiments?

Response: *Apologies, this is now done on line 282 in the main text. We have also added this information to the figure legend for Figure 3g and h.*

Referee #2 - comparative genomics, crops (Remarks to the Author):

Summary of key results: In this manuscript, the authors use single-cell RNAseq combined with ATAC-seq to determine the mechanism behind the transcriptional re-wiring that was key in the evolution of C4 photosynthesis. They find that the C4 plant sorghum co-opted an existing regulatory network in order to activate photosynthesis genes within the bundle sheath cells, which was an essential innovation for this more efficient form of photosynthesis.

I found this manuscript very interesting, with elegant experimental design and an exciting and scientifically robust result that offers insight into one of the most important biological pathways in plants. The methods used were well-suited to study the question presented by the authors, the analyses were appropriate, and the results were all presented very clearly in the figures.

I do think that the authors could do more to make their manuscript more accessible to non-plant scientists, who may have trouble recognizing the significance of the final result if they don't fully understand how the localization of photosynthesis to different cell types is important. The authors do discuss this idea in the introduction, but my suggestion would be to also add a figure to the beginning of the manuscript giving a broad overview of C3 versus C4 photosynthesis, to help orient readers who may be much less familiar with this process. This would provide a nice set-up to Figure 5D, where the main findings are summarized.

Response: *We thank the reviewer for this positive analysis of our work. We acknowledge the point regarding accessibility for non-specialists and have now included an additional panel as suggested in Figure 1, and refer to this on line 81.*

I also have a few other minor comments/questions:

1. In lines 101-102, it is mentioned that sorghum and rice may share molecular signatures with the many other species derived from their last common ancestor. There's a nice paper from back in 2010 that did ancestral karyotype reconstruction which shows how much of the ancestral genome (and genome structure) both sorghum and rice still share with each other and the grass ancestor, which I think lends even more weight to this argument that these species are an excellent model for the work being done here. (Murat et al. 2010. Ancestral grass karyotype reconstruction unravels new mechanisms of genome shuffling as a source of plant evolution. *Genome Research* 20: 1545-1557.)

Response: *Thank you very much for highlighting this point, and this paper. We agree and have now made reference to this work on lines 101-104.*

2. In lines 211-214: The phrase “gained genes” makes it sound like sorghum gained new gene copies or novel gene sequences, but I don't think that is what you mean since (as I understand it) you are only looked at expression in genes with orthologs in rice. This should be clarified.

Response: *This is true, apologies for the poorly written text. We have now revised this paragraph (lines 218-221) to clarify. Specifically, we now write:*

“Thus, to transcriptionally rewire the C₄ bundle sheath it appears that this cell type (i) gained cell-type specific expression of genes not preferentially or highly expressed in rice shoot tissue, (ii) lost cell-type specific expression of genes specifically transcribed in the C₃ bundle sheath, and (iii) gained cell-type specific expression of genes preferentially expressed in other cell types of C₃ rice including mesophyll and guard cells.

3. In line 573 of the Methods, it is stated that “if multiple orthologs for a gene were found across species, only one was retained.” Does this mean that you also removed any expression data if transcripts mapped to a duplicate gene copy, and only retained counts for one copy? Is this definitely controlling for the possibility that some increases in expression could come from copy number variants (and not just gene regulation)?

Response: Thank you for raising this important point. In our generation of the pan-transcriptome atlas, we used 1:1 ortholog conversion, and thus ignored additional orthologs if they were present. We believe that this is the most straight forward way to generate such an atlas; it cannot be achieved with multiple ortholog pairings. However, in our subsequent analysis of conservation of overlapping cell type specific markers between species (i.e. Figure 2C – D), and our assessment of ortholog gene pairs across each cell type pair (Figure 2E – 2G), we used orthogroups as a means to compare expression across species. This means that if a rice gene had 2 sorghum orthologs (for example), both sorghum orthologs were included in the analysis. We apologize that this was not sufficiently clear and have now added additional detail to articulate this point further (lines 207 and 235-237).

We note that Reviewer 4's comment 1 was similar. To meet Reviewer 4's request, we also performed an additional analysis that investigates whether higher order combinations of orthologous groups are partitioned in their expression across the mesophyll and bundle sheath of rice and sorghum. This analysis is presented in Extended Data Figure 6, and indicated that only 12% of orthologous gene pairs show disparate partitioning patterns.

Referee #3 - evolution of photosynthesis (Remarks to the Author):

This is an important and exciting study that is well designed, well executed, and well described. Though some technical elements were outside my area of expertise, I nonetheless was able to follow along quite well due to their clear explanations. Their results are important for our understanding of C4 photosynthesis evolution, but also evolutionary innovation more broadly.

The basic question this paper asks is how, during C4 evolution, the bundle sheath cells become recruited as the major carbon-fixing cells of the plant leaf. Through analyses of single-cell RNA expression of sorghum (C4) and rice (C3), they discover that the core set of Calvin-cycle genes as well as a key decarboxylation gene have acquired key cis-element motifs in Sorghum that are part of a bundle-sheath identity network, regulated by a DOF family transcription factor. Their inference is that this cell-identity network is ancestral, because it also regulates bundle sheath development in rice. Thus, sorghum utilized a pre-existing bundle sheath identity module to incorporate cell-specific expression of key photosynthesis genes that is essential for a functional C4 cycle. They contrast this result to other papers focused in eudicots, which suggest that the expression of the transcription factors themselves change spatially, and cis-elements are conserved.

Response: We thank the reviewer for their careful and positive analysis of our work.

One important contribution from this paper is the identification of bundle-sheath (BS) specific marker genes in C3 plants- this apparently required transformation and fluorescent labeling of BS nuclei with a promoter of phosphoenolpyruvate carboxykinase (PCK). This is not my area of expertise, but as I thought more about it, it became unclear to me why they took this additional step, as they are simply sorting nuclei according to PCK expression. So they are already using PCK as a "marker gene" of C3 BS, based on Nomura et al. 2005. Wouldn't they get the same result if they had used PCK as a marker gene in the same way they used other known localized genes in mesophyll, cuticle, xylem, etc, i.e., without doing the transformation experiment? The rationale could be better explained here.

Response: This is a good point, and we apologise this was not presented clearly enough in the first version of the manuscript. The PCK promoter that drives bundle sheath expression in rice comes originally from the C₄ plant *Zoysia japonica*, and so is not an "endogenous" C₃ bundle sheath marker, rather the promoter drives a transgene reporter that Nomura et al 2005 had shown generated clear expression in the rice bundle sheath. In fact, the reason we took this approach was that it was very difficult to identify the bundle sheath cluster in rice from endogenous patterns of expression as there were so few markers available. We have now inserted the following text at lines 155-158 to clarify this. "We therefore generated a stable reporter line in which bundle sheath nuclei were labelled with a fluorescent mTurquoise2 reporter under control of the PHOSPHOENOLCARBOXYKINASE promoter from the C₄ plant *Zoysia japonica*²⁷ (Fig. 1e)."

My only concern with the paper might be an over-confidence in what is inferred to be "ancestral", as the experiment with sorghum and rice is really a two-taxon statement – there is literally no way to infer any directionality to differences found between two species. The authors infer that the sorghum state is derived because we know that C4 is derived – it is not terrible logic, but they should carefully

lay out their reasoning for the readers. One resource that is readily available to them is a good sampling of annotated Poaceae genomes accessible via Phytozome- including important C3 PACMAD species like *Chasmanthium*. While not nearly as informative as the ATAC-seq analyses performed in their experiments, the authors could very quickly target some of their BS-specific genes and examine upstream regions for DOF-binding motif enrichment across a broader collection of grasses. I'm not insistent but I think a quick analysis along these lines would lend support to their inferences of the DOF-BS identity network being ancestral.

*Response: We agree that it is not possible to infer ancestral state from a sampling of two. We have rephrased the text where appropriate indicating that it is the C₃ state that is considered ancestral to the C₄ state. As suggested, we have also carried out cis-element enrichment analyses in homologs of rice and sorghum genes consistently partitioned to the bundle sheath in *Chasmanthium laxum* (as a member of the PACMAD clade), *Brachypodium distachyon*, and *Hordeum vulgare*. These analyses indicate that DOF motifs are enriched in homologs of BS-specific genes in a range of different Poaceae species from the PACMAD and also the BEP lineages. We have added these results in Extended Data Figure 12/Supplementary Table 11 and have described the findings on lines 327-330. In addition, we have made reference to these findings in the discussion on lines 403-406.*

Smaller point about their title- is this really a 're-wiring'? It seems the opposite to me- conservation of an old wiring, but adding in new components.

Response: We thank the reviewer for pointing this out. We have now retitled our manuscript "Exaptation of ancestral cell identity networks enables C₄ photosynthesis"

Referee #4 plant genomics, bioinformatics (Remarks to the Author):

This manuscript describes the generation of complementary high-quality datasets in two cereal species, rice and sorghum, to study the molecular signaling cascades controlling C3 and C4 photosynthesis. Based on high-resolution transcript and accessible chromatin profiling using state-of-the-art single-cell -omics methods, gene expression and chromatin state dynamics are characterized for seedlings undergoing photomorphogenesis. Comparative analysis using known and novel marker genes is used to annotate different clusters, representing distinct cell types (incl. validation of a bundle sheath reporter line). Comparative analysis using gene orthology information is subsequently used to characterize similarities and differences in gene expression, focusing on mesophyll and bundle sheath, two cell types playing a key role in both photosynthesis pathways. The analysis of expression divergence between both species allows the delineation of a set of genes showing differential partitioning, very likely contributing to the differences between the C3 and C4 pathways. Chromatin accessibility analysis during dark and light confirms major differences in the regulation of photosynthesis genes, suggesting that cell identity and associated cis-elements are responsible for the partitioning of photosynthesis genes. Taken together, the authors propose a model where a major difference between rice (C3) and sorghum (C4) is the acquisition of DOF cis-elements in sorghum photosynthesis genes expressed in bundle sheath.

The manuscript starts with an excellent introduction outlining the relevance and major differences between C3 and C4 photosynthesis in plants. The transcript and accessible chromatin profiling results are well described. The Figures and Extended Data Figures do a good job to report the major trends in this complex cross-species dataset, though on some occasions the figure legends are very brief to fully understand all results shown in the different panels. While I largely agree with the interpretation of the reported gene expression differences between rice and sorghum, I am missing detailed results on the cis-element analysis that fully support the claims that are made in the proposed model. Furthermore, and in contrast to previous reports by the same authors, no validation experiments were performed to test some of the hypotheses that are resulting from the proposed model.

Response: We thank the reviewer for their careful and positive analysis of our work. Below we outline the changes we have made to the manuscript in response to their excellent analysis.

Major

1. The authors rely on gene orthology information to perform various comparative analyses between rice and sorghum. Line 572 mentions ortholog conversions were performed in a one-to-one manner, meaning that in some cases, when multiple orthologs are present, only one ortholog was retained.

While I acknowledge that for some comparative analyses orthogroups are used, it is unclear how cases where multiple orthologs (e.g. one rice gene has two sorghum orthologs) showing different or opposite expression patterns (e.g. up vs down, or BS vs M for the two sorghum orthologs) are dealt with, and how these affect the presented partitioning results. Please provide more details incl. counts on the number of one-to-one and more complex orthology relationships.

Response: *We agree, we also address this in our reply to Reviewer 2 point 3 above. Our descriptions were not clear enough here. We have added a more detailed description of the analyses that were performed for each comparison in the methods section "Orthology analyses". We have also performed an additional analysis that investigates how higher order combinations of orthologous groups are partitioned in their expression across the mesophyll and bundle sheath of rice and sorghum. This analysis is now presented in Extended Data Figure 6, and indicates that more complex ortholog pairs showing disparate gene expression patterns are relatively rare - occurring in around 12% of cases. We have referred to this analysis in the main text on lines 235-237.*

2a. Results reported in Suppl. Tables 10-12 (please add TF family information for these motif identifiers) are insufficient to understand how different TF motifs are in both cell types, for both species.

Response: *Thank you for this point. We now add transcription factor family information for each motif presented in this spreadsheet.*

2b. To verify the important claims made in the article about the role of DOF TF motifs in genes swapping expression (lines 338-339), the authors need to report motif frequencies for the different gene sets (Fig 5a) in rice and sorghum. The adjusted p-values reported in the Suppl. Tables are insufficient to verify how many genes, and which genes, have specific TF motifs.

Response: *Thank you for this point, we agree. We now report motif frequencies for genes presented in Figure 5a in a dotplot (Figure 5b). This analysis was performed by running FIMO on open chromatin within +/- 1500 base pairs of the closest transcriptional start site. This analysis shows that enrichment of DOF motifs (i.e. a larger number of DOF motifs in sorghum orthologs compared with rice orthologs) was observed for the majority of the genes analysed. This enrichment is not seen when we quantify the number of DOF motifs in genes that are consistently partitioned to the bundle sheath in rice and sorghum. We have added these additional analyses in Figure 5b, Extended Data Fig. 14, and Supplementary Table 13, and have described the findings in the main text on lines 355-359.*

2c. The example shown in Fig 5b is not very convincing (the GAPDH and ALDOLASE peaks contain DOF motifs in both rice and sorghum, despite the difference in accessible chromatin; the NADP-ME example is better).

Response: *Thank you for pointing this out. We agree that the previous version was not as compelling as it could have been. In looking into this, we found that our previous approach had introduced redundant information on motif frequency because we had used the consensus site associated with the whole DOF transcription factor family. However, the DOF core motif that is common to all the DOF binding sites consists of the sequence AAAG, which corresponds most closely to the Dof2 binding site. Therefore, identifying this core motif in accessible chromatin regions also covers the core of other variations of the DOF motif. We have now refined our analysis to looking at the DOF core motif, which clearly shows that the number of DOF motifs in open chromatin of sorghum orthologs is larger than in the corresponding rice orthologs. We now show GAPDH as an example of this enrichment in Fig. 5c, where the number of DOF motifs increases from 4 in rice to 9 in sorghum.*

2d. Furthermore, a limitation of the current study is that no evidence is presented which DOF TFs are controlling BS expression (see comment 4).

Response: *This is a good point. We have now used 4 bundle-sheath preferential DOF transcription factors from rice and 4 bundle-sheath preferential DOF transcription factors from sorghum as effectors in transactivation assays in rice protoplasts. These experiments show that several DOFs*

are sufficient to activate expression of sorghum GAPDH and NADP-ME as well as the rice minimal SIR promoter, and thus support the model in which DOF transcription factors are acting to regulate genes expressed in the bundle sheath. We describe these new findings on lines 369-377.

3. Following the previous comment, based on the difference in accessible chromatin shown in Extended Data Figure 13a (the BS accessibility in rice is much lower than in sorghum), a difference in the control of accessible chromatin (without a direct role of specific TFs, so potentially epigenetic) could provide an alternative explanation. Validation experiments, for example by perturbation of specific cis-elements to measure the effect on gene expression and/or photosynthetic activity, like the authors did in Singh et al., 2023, are needed to strengthen the proposed model.

Response: Thank you, this is also an excellent point. We agree with the reviewer that changes in accessible chromatin could play a role in changing cell type specific expression of photosynthesis genes. As suggested, we now present new data from stable transgenic lines of rice in which the DOF motifs have been mutated in a promoter that drives expression in the bundle sheath. When the DOF binding sites were altered, we detected a statistically significant reduction in expression. These data confirm that the DOF motif does control expression in the rice bundle sheath, and thus provides further support for the model in which gain of DOF motifs by C₄ genes allows their expression in bundle sheath cells to be amplified. Given that eukaryotic promoter systems contain complex networks of transcription factor binding sites, this is not unexpected. These findings are presented in Figure 5 and discussed on lines 377-380.

4. I suggest the authors include reference <https://pubmed.ncbi.nlm.nih.gov/34783109/> in the Discussion, as this paper performed related experiments in maize, also hinting towards the role of DOF TFs incl. ChIA-PET data linking specific TFs to photosynthesis genes.

Response: Thank you, this is also a good point. We now include this in the discussion (lines 399-400).

5. Several figures have a very brief legend, making it hard to fully understand the different panels.

Response: We apologise that the legends were not sufficiently comprehensive. We have now added additional details and explanations to the figure legends.

a. Fig 1C. transcript profiles of which assay are shown here (cfr. Fig1A)?

Response: Apologies for the lack of clarity on this. We have now updated the figure panel legend to read:

“(d) rice and (e) sorghum single nuclei, encompassing all time points assayed by 10X RNA-seq and sci-RNA-seq3.”

b. Fig 2f: please extend text legend (too cryptic to understand)

Response: We have updated this figure legend to read: “Overlap of genes found differentially partitioned between each cell-type pair in rice and sorghum. The agreement between mesophyll and bundle sheath partitioned genes across species is shown on the left, and the same analysis is repeated for all other possible cell type pairs (right).”

c. Fig 2g: add gene symbols of important or relevant genes (suggestion). Or refer to the corresponding Suppl. Table where these genes are listed.

Response: Thank you for this suggestion, we now refer to the corresponding Supplementary Table (Supplementary Table 6) in the text on line 235 and have added gene descriptions to the Supplementary Table.

d. Extended data Figures 4, 6: please add cell type annotation for the different clusters. For example, in Extended Data Figure 4, add that cluster 10 is BS.

Response: We have now added the annotation for the bundle sheath cluster in both figures.

e. Extended Data Figure 7: please check the legend, there is an error in part (c).

Response: This has now been corrected.

6. Fig 4e and Extended Data Figure 11-12: unclear which motifs are specific for which cell type in which species. Why are the top 25 motifs compared (do the conclusions change when comparing more/all enriched motifs)?

Response: *We apologise that this extended data was not explained fully. We used the Signac workflow to call cell-type specific enriched cis-regulatory elements (see github code Script 51 and 57). This workflow draws upon the JASPAR database of cis-regulatory elements for all plants, of which there are 530 in total. We found that this statistical approach delivered many significantly enriched cis-regulatory elements per cell type. For example, in the rice mesophyll, this approach reported 144 significant motifs, or 27% of the known motifs. We considered this too broad a list – and thus decided to institute a second threshold – i.e. taking the top 25 most significant. This is now explained more clearly in the methods on lines 722-731.*

7. For specific photosynthetic gene sets discussed in the article, please provide gene – TF motif info (in accessible peaks), as apart from the three genes discussed in the text, no info for other photosynthesis genes is given.

Response: *We have added gene names for photosynthesis genes in the heatmap from Fig. 5a, and have also provided a new Supplementary Table 13 that indicates the names of genes in Figure 5A as well as the Dof2 motif frequencies in orthologs of rice and sorghum.*

Minor

8. Line 329-330: Given the previous sentences reported specific genes, the start of this sentence does not make sense. I assume you want to identify motifs associated with (differential?) accessible chromatin peaks for these genes? Please revise.

Response: *We thank the reviewer for reporting this inconsistency. We have revised the paragraph by simply removing this sentence. We find this improves the logical flow. Specifically, on lines 341-345 we now write: “Among the 40 orthologs in this category were the Calvin-Benson-Bassham cycle genes FRUCTOSE BISPHOSPHATE ALDOLASE and GLYCERALDEHYDE 3-PHOSPHATE DEHYDROGENASE (GAPDH), photorespiration genes such as GLYCOLATE OXIDASE, and light reaction genes including the LHClI subunit. Strikingly, among these differentially partitioned genes, we found that associated chromatin was enriched in cell-type specific Myb-related, High mobility group ... “*

8a. Line 335: move to Methods.

Response: *This change has been made.*

8b. As permutation statistics were applied here, how was the statistical significance of enriched motifs determined for the previously reported motif results. Why were different methods used?

Response:

We used two different approaches to identify enriched cis-elements: The first one was to find cis-regulatory elements that were enriched within a cell type (Figure 4D). For this, we looked holistically at all accessible chromatin within a cell type cluster without considering whether peaks were associated with a particular gene set. This approach was implemented using the chromVAR function in Signac. We have improved our description of this approach in the Methods section (lines 722-731), and this is displayed in Extended Data Figure 11.

The second approach we used was designed to investigate specific genes rather than genome-wide responses. Because we wanted to understand cis-regulatory enrichment within peaks proximal to differentially partitioned genes, we needed to take a different statistical approach that only examined this gene set. To this end we called enriched motifs among differentially partitioned genes (Figure 5A) using the FindMotifs() command from Signac. This is a hypergeometric test that compares accessible chromatin to a randomized genomic background. Because this background is a random sample, we wanted to ensure that the motifs we computed were robust and so tested this across 100 permutations. We then computed each motif's average rank across these 100 permutations. This approach has been clarified in the Methods section (lines 735-743) and the ranks are presented in the updated Supplementary Table 13.

8c. The Methods section on Cis-element analyses is hard to understand (also see next comment) and hard to reproduce.

Response: *We apologise for the lack of clarity here. We have now revised the methods section to address this.*

8d. Sharing source code or example command lines how these tools were executed, on which datasets, would help understanding what was done.

Response: *We have now uploaded all R code related to this manuscript to github (<https://github.com/joey1463/C3-C4/tree/main>). Code specific for this analysis can be found in Script 63 (for rice) and Script 68 (for sorghum).*

9. Line 622: Unclear what you mean with “we calculated position frequency matrices’. You converted PWMs to position frequency matrices? Or you mapped known JASPAR (I assume JASPER is a typo) to the rice and sorghum genome? Or you did de novo motif finding to learn position frequency matrices? Please revise.

Response: *Apologies. In our revised manuscript we have now clarified text explaining this in the methods. We relied on position weighted matrices as found in the JASPAR database - we did not compute motifs de novo.*

10. Line 639: “ignoring areas that were inside gene bodies.” Does this mean that cis-regulatory elements in introns or UTR are all discarded? Given the increasing evidence that also these gene regions are important in controlling gene expression, this seems like a harsh filtering step.

Response: *We agree, this approach was more conservative than was sensible given the importance of regulation within genes. We have revised our analysis and text in light of this comment. In our new Figure 5C, we now search for DOF motifs within peaks that are accessible within +/- 1500bp of the transcriptional start site, and so if accessible peaks were detected in gene bodies, these are now included. This analysis approach has been updated in the methods on lines 744-749: “To quantify the occurrence of DOF binding sites, we extracted the genomic sequence of peaks that were proximal to the transcription start site (+/- 1500 bp). If a peak was proximal to two transcription start sites it was assigned to the closer one. We then implemented Find Individual Motif Occurrences (FIMO) to quantify the number of DOF consensus sites within these chromatin regions (p -value threshold = 0.005). We chose the DOF2 (MA0020.1) motif as representative of the core DOF consensus sequence AAAG.”*

Reviewer Reports on the First Revision:

Referees' comments:

Referee #1 (Remarks to the Author):

The authors have addressed all of my concerns. This is an exciting addition to the field.

Referee #2 (Remarks to the Author):

In the revised version of their manuscript, the authors have sufficiently addressed all of my questions and comments from my original review. In particular, they have provided a lot of clarity to their methods and their figures. I do not have any additional comments.

Referee #2 (Remarks on code availability):

I have checked the Github page provided to ensure that the code described by the authors was in fact available. I can confirm that all of the scripts are available, they are very nicely labeled in a numerical order to make it more clear which analyses they were used for in the manuscript, and the README page provides details on what each script is and what it produces.

Referee #4 (Remarks to the Author):

I thank the authors for taking into consideration my comments and suggestions formulated on the initial submission. The revised manuscript now gives additional information about the cis-elements/TFBS signals present in the different datasets, supporting the claims made in the paper. Furthermore, the experimental validations start addressing some of the hypotheses formulated based on the multi-omics data analysis. I am looking forward to seeing this paper published.

Comments:

- The Methods section writes "R scripts used for analyses presented in this manuscript are published on github at <https://github.com/joey1463/C3-C4/>, and associated files can be found on Mendeley.", but where on Mendeley can these be found?

Referee #4 (Remarks on code availability):

Whereas a README gives a brief overview of the different R scripts used for data analysis, several of these scripts load Rdata files (e.g., script 51, `load("L1_rice_multiome_RNA_combined_with_ATAC.RData")`). Are these Rdata files available? The Methods section writes "R scripts used for analyses presented in this manuscript are published on github at <https://github.com/joey1463/C3-C4/>, and associated files can be found on Mendeley.", but where on Mendeley can these be found?

Author Rebuttals to First Revision:

Referee #1 (Remarks to the Author):

The authors have addressed all of my concerns. This is an exciting addition to the field.

We thank the reviewer for their input.

Referee #2 (Remarks to the Author):

In the revised version of their manuscript, the authors have sufficiently addressed all of my questions and comments from my original review. In particular, they have provided a lot of clarity to their methods and their figures. I do not have any additional comments.

We thank the reviewer for their input.

Referee #2 (Remarks on code availability):

I have checked the Github page provided to ensure that the code described by the authors was in fact available. I can confirm that all of the scripts are available, they are very nicely labeled in a numerical order to make it more clear which analyses they were used for in the manuscript, and the README page provides details on what each script is and what it produces.

We thank the reviewer for checking our Github page.

Referee #4 (Remarks to the Author):

I thank the authors for taking into consideration my comments and suggestions formulated on the initial submission. The revised manuscript now gives additional information about the cis-elements/TFBS signals present in the different datasets, supporting the claims made in the paper. Furthermore, the experimental validations start addressing some of the hypotheses formulated based on the multi-omics data analysis. I am looking forward to seeing this paper published.

We thank the reviewer for their input.

Comments:

- The Methods section writes "R scripts used for analyses presented in this manuscript are

published on github at <https://github.com/joey1463/C3-C4/>, and associated files can be found on Mendeley.", but where on Mendeley can these be found?

In our revised submission, we have clarified this point. The associated files for R analysis (i.e. the Seurat atlas files) can be downloaded directly from the GEO database (GSE248919). Datasets published on Mendeley consist of raw data related to non-genomic assays (i.e. chlorophyll quantification, transactivation assays, and GUS reporter line quantification).

Referee #4 (Remarks on code availability):

Whereas a README gives a brief overview of the different R scripts used for data analysis, several of these scripts load Rdata files (e.g., script 51, "load("L1_rice_multiome_RNA_combined_with_ATAC.RData)"). Are these Rdata files available?

This relates to the comment above. The file "L1_rice_multiome_RNA_combined_with_ATAC.RData", alongside other clustered data, can be found at GEO database (GSE248919).